# Two bifunctional inositol pyrophosphate kinases/phosphatases control plant phosphate homeostasis

Jinsheng Zhu[1], Kelvin Lau[1†], Robert Puschmann[2,3], Robert K Harmel[2,3], Youjun Zhang[4,5], Verena Pries[6], Philipp Gaugler[6], Larissa Broger[1], Amit K Dutta[7], Henning J Jessen[7], Gabriel Schaaf[6], Alisdair R Fernie[4], Ludwig A Hothorn[8‡], Dorothea Fiedler[2,3], Michael Hothorn[1*]

[1]Structural Plant Biology Laboratory, Department of Botany and Plant Biology, University of Geneva, Geneva, Switzerland; [2]Leibniz-Forschungsinstitut für Molekulare Pharmakologie, Berlin, Germany; [3]Department of Chemistry, Humboldt Universität zu Berlin, Berlin, Germany; [4]Max-Planck Institute of Molecular Plant Physiology, Potsdam-Golm, Germany; [5]Center of Plant System Biology and Biotechnology, Plovdiv, Bulgaria; [6]Institute of Crop Science and Resource Conservation, Department of Plant Nutrition, University of Bonn, Bonn, Germany; [7]Institute of Organic Chemistry, Freiburg im Breisgau, Germany; [8]Institute of Biostatistics, Leibniz University, Hannover, Germany

*For correspondence:
michael.hothorn@unige.ch

‡retired

Present address: †Protein Production and Structure Core Facility, École Polytechnique Fédérale de Lausanne, Lausanne, Switzerland

Competing interests: The authors declare that no competing interests exist.

**Abstract** Many eukaryotic proteins regulating phosphate (Pi) homeostasis contain SPX domains that are receptors for inositol pyrophosphates (PP-InsP), suggesting that PP-InsPs may regulate Pi homeostasis. Here we report that deletion of two diphosphoinositol pentakisphosphate kinases VIH1/2 impairs plant growth and leads to constitutive Pi starvation responses. Deletion of phosphate starvation response transcription factors partially rescues vih1 vih2 mutant phenotypes, placing diphosphoinositol pentakisphosphate kinases in plant Pi signal transduction cascades. VIH1/2 are bifunctional enzymes able to generate and break-down PP-InsPs. Mutations in the kinase active site lead to increased Pi levels and constitutive Pi starvation responses. ATP levels change significantly in different Pi growth conditions. ATP-Mg2+ concentrations shift the relative kinase and phosphatase activities of diphosphoinositol pentakisphosphate kinases in vitro. Pi inhibits the phosphatase activity of the enzyme. Thus, VIH1 and VIH2 relay changes in cellular ATP and Pi concentrations to changes in PP-InsP levels, allowing plants to maintain sufficient Pi levels.
DOI: https://doi.org/10.7554/eLife.43582.001

## Introduction

Phosphorus is a growth limiting nutrient for plants, taken up from the soil as inorganic phosphate ($H_2PO_4^-$, Pi). Plants can also take up phosphite (Phi), a reduced form of Pi which however cannot be used as a source of phosphorus (*Ticconi et al., 2001*). Plants and other soil-living eukaryotes have evolved sophisticated Pi sensing, uptake, transport and storage mechanisms (*Puga et al., 2017*). Many of the proteins involved in these processes contain small hydrophilic SPX domains (*Secco et al., 2012*). We have previously shown that these domains act as cellular receptors for inositol pyrophosphate (PP-InsP) signaling molecules (*Wild et al., 2016*). PP-InsPs are composed of a fully phosphorylated inositol ring containing one or two pyrophosphate moieties (*Shears, 2018*). PP-InsPs bind to a basic surface cluster highly conserved among SPX domains to regulate divers biochemical and cellular processes in fungi, plants and animals: The N-terminal SPX domains of the yeast VTC

complex stimulates its inorganic polyphosphate polymerase activity when bound to 5PP-InsP$_5$ (*Hothorn et al., 2009*; *Wild et al., 2016*; *Gerasimaite et al., 2017*). In plant and human phosphate exporters, mutations in the PP-InsP binding surfaces of their N-terminal SPX domains affect their Pi transport activities (*Legati et al., 2015*; *Wild et al., 2016*). Plants contain additional soluble, stand-alone SPX proteins, which have been shown to bind PHOSPHATE STARVATION RESPONSE (PHR) transcription factors (*Rubio et al., 2001*; *Puga et al., 2014*; *Wang et al., 2014*; *Qi et al., 2017*). PHR1 and its homolog PHL1 are master regulators of the plant Pi starvation response, which enable plants to grow and survive in Pi limiting growth conditions (*Bustos et al., 2010*). PP-InsP-bound SPX domains can physically interact with PHR1, keeping it from binding DNA (*Puga et al., 2014*; *Wang et al., 2014*; *Wild et al., 2016*; *Qi et al., 2017*). In the absence of PP-InsPs, the SPX – PHR1 complex dissociates and PHR1 oligomers can target the promoters of Pi starvation induced (PSI) genes, resulting in major changes in plant metabolism and development (*Wild et al., 2016*; *Jung et al., 2018*). These findings suggest that changes in cellular PP-InsP levels may regulate eukaryotic Pi homeostasis. 5PP-InsP$_5$ levels are indeed reduced in yeast and in animal cells grown in Pi starvation conditions (*Lonetti et al., 2011*; *Wild et al., 2016*; *Gu et al., 2017a*), suggesting that PP-InsP metabolic enzymes may be key regulators of Pi homeostasis (*Gu et al., 2017a*; *Azevedo and Saiardi, 2017*; *Shears, 2018*).

PP-InsP biosynthesis is well understood in yeast and mammals, where inositol hexakisphosphate kinases Kcs1/IP6K catalyze the synthesis of 5PP-InsP$_5$ from InsP$_6$. 5PP-InsP$_5$ can be further phosphorylated by diphosphoinositol pentakisphosphate kinases Vip1/PPIP5K to 1,5(PP)$_2$-InsP$_4$ (referred to as InsP$_8$ in this study) (*Shears, 2018*) (*Figure 1—figure supplement 1*). How these enzymes may sense changes in cellular Pi concentration remains to be elucidated in mechanistic detail. Currently, it is known that Kcs1/IP6Ks have a K$_m$ for ATP in the low millimolar range and may therefore generate 5PP-InsP$_5$ in response to changes in cellular ATP levels (*Saiardi et al., 1999*; *Gu et al., 2017a*; *Gu et al., 2017b*). In addition, human PPIP5Ks can be regulated by Pi itself, stimulating their 5PP-InsP$_5$ kinase activity, and inhibiting their InsP$_8$ phosphatase activity (*Gu et al., 2017a*). Vip1/PPIP5Ks are bifunctional enzymes harboring an N-terminal InsP kinase and a C-terminal phosphatase domain (*Figure 1A*) (*Mulugu et al., 2007*; *Fridy et al., 2007*; *Wang et al., 2015*). The yeast and animal kinase domain acts on the phosphate group at the C1 position of the inositol ring of both InsP$_6$ and 5PP-InsP$_5$, yielding 1PP-InsP$_5$ or InsP$_8$ reaction products (*Mulugu et al., 2007*; *Fridy et al., 2007*). The phosphatase domain has been characterized as a specific 1PP-InsP$_5$ / InsP$_8$ phosphatase in yeast (*Wang et al., 2015*). Deletion of PPIP5K in human cells results in lower InsP$_8$ and elevated ATP levels, due to enhanced mitochondrial oxidative phosphorylation and increased glycolysis (*Gu et al., 2017b*).

In plants, no IP6K enzyme has been identified thus far, but Vip1/PPIP5K orthologs Vip1 and VIH1/2 have been reported from Chlamydomonas (*Couso et al., 2016*) and Arabidopsis (*Desai et al., 2014*; *Laha et al., 2015*), respectively. The algal and plant enzymes share the bifunctional kinase/phosphatase domain architecture with yeast and animal PPIP5Ks. Arabidopsis VIH1 and VIH2 can rescue yeast *vip1* but not *ksc1* mutants (*Desai et al., 2014*; *Laha et al., 2015*). Deletion of the Chlamydomonas Vip1 results in nutrient signaling phenotypes affecting carbon metabolism (*Couso et al., 2016*). The Arabidopsis *vih2-4* loss-of-function mutant has lower InsP$_8$ but increased InsP$_7$ levels, suggesting that the enzyme may generate an InsP$_8$ isoform in vivo (*Laha et al., 2015*). *VIH2* mutant lines have been reported to affect jasmonic acid-regulated herbivore resistance, as PP-InsPs also form critical co-factors of the jasmonate receptor complex (*Sheard et al., 2010*; *Laha et al., 2016*). No obvious Pi starvation phenotype has been reported for these mutants (*Kuo et al., 2018*). To analyze if indeed plant Pi homeostasis is mediated by PP-InsPs, we characterized Pi signaling related phenotypes of *vih1* and *vih2* mutants in Arabidopsis and dissected the individual contributions of their conserved InsP kinase and phosphatase domains to Pi homeostasis, Pi starvation responses, and to the signaling capacity of Phi.

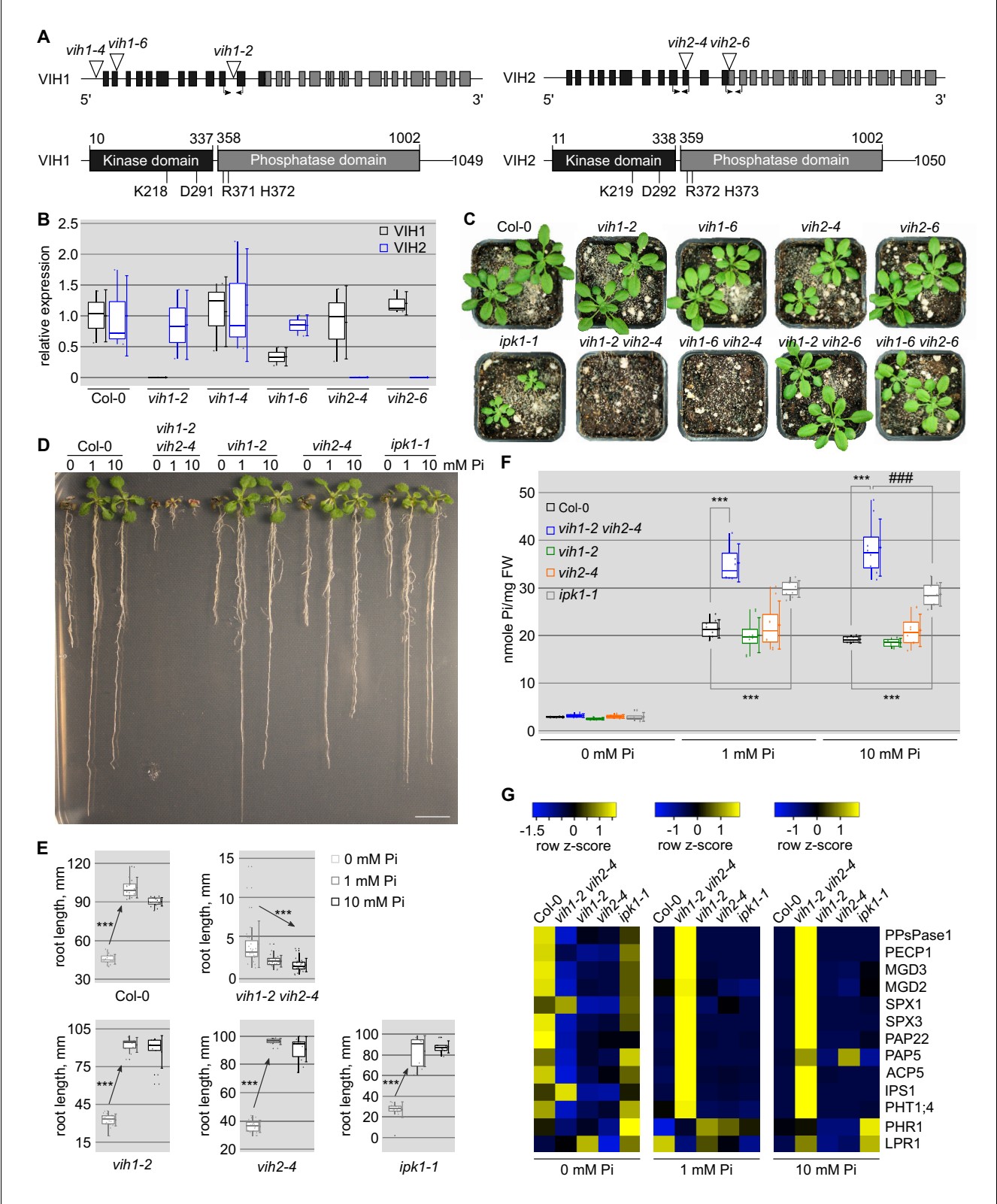

**Figure 1.** *vih1 vih2* loss-of-function mutants show severe growth phenotypes and hyperaccumulate Pi. (**A**) Schematic overview of VIH1 and VIH2: (upper panel) VIH1 and VIH2 genes with exons described as rectangles, introns as lines. T-DNA insertions are depicted as triangles, primer positions used in qRT-PCR analyses are indicated by arrows. (lower panel) VIH1 and VIH2 protein architecture, with kinase domains shown in black, phosphatase domains in gray and putative linkers/unstructured regions as lines. The point mutations used in this study are shown below the domain schemes. (**B**) qRT-PCR

*Figure 1 continued on next page*

*Figure 1 continued*

expression analysis of VIH1 and VIH2 transcripts in the T-DNA mutant allele backgrounds. Shown are $2^{-\Delta CT}$ values relative to Col-0 wild-type. Quantifications were done using three biological replicates. (C) Growth phenotypes of *vih1* and *vih2* single mutant, and of *vih1 vih2* double mutants. Shown are plants 20 DAG in soil compared to Col-0 wild-type. (D) Growth phenotypes of *vih1-2 vih2-4*, *vih1-2*, *vih2-4*, and of *ipk1-1* compared to Col-0 wild-type seedlings. Plants were germinated in vertical $^{1/2}$MS plates for 8 d, transferred to Pi-deficient $^{1/2}$MS plates supplemented with either 0 mM, 1 mM or 10 mM Pi and grown for additional 6 d. Scale bars correspond to 2 cm. (E) Trend analysis of seedling root length vs. cellular Pi concentration for the seedlings described in (D). For each boxed position, root length measurements were performed for seedlings from three independent MS plates. (F) Pi contents of the seedlings shown in (D) 14 DAG. For each boxed position, four independent plants were measured with two technical replicates. (G) qRT-PCR quantification of PSI marker genes (PPsPase1, PECP1, MGD3, MGD2, SPX1, SPX3, PAP22, PAP5, ACP5, IPS1, PHT1;4) and of the non-PSI genes PHR1 and LPR1 in Col-0 wild-type, *vih1-2 vih2-4*, *vih1-2*, *vih2-4* and *ipk1-1* seedlings described in (D). Expression levels are represented as Z-scores. The original data are shown in *Supplementary file 3a*.

DOI: https://doi.org/10.7554/eLife.43582.002

The following figure supplements are available for figure 1:

**Figure supplement 1.** Overview of PP-InsP isoforms and kinases involved in inositol pyrophosphate metabolism.
DOI: https://doi.org/10.7554/eLife.43582.003

**Figure supplement 2.** The kinase domain and phosphatase domain of VIHs/PPIP5Ks are conserved among different species.
DOI: https://doi.org/10.7554/eLife.43582.004

**Figure supplement 3.** VIH1 and VIH2 show partly unique and partially overlapping expression patterns.
DOI: https://doi.org/10.7554/eLife.43582.005

**Figure supplement 4.** VIH1 and VIH2 are cytoplasmic enzymes with partially overlapping expression domains.
DOI: https://doi.org/10.7554/eLife.43582.006

**Figure supplement 5.** Inducible knock-down of VIH1 in the *vih2-4* background impairs plant growth and leads to shoot Pi accumulation.
DOI: https://doi.org/10.7554/eLife.43582.007

## Results

### Deletion of VIH1/VIH2 affects plant growth and phosphate homeostasis

Arabidopsis VIH1 and VIH2 show 89% sequence identity at the protein level (*Figure 1—figure supplement 2*). VIH1 has been previously reported to be specifically expressed in pollen while VIH2 showed a broad expression pattern as judged from qRT-PCR experiments (*Desai et al., 2014*; *Laha et al., 2015*). We further analyzed *VIH1* and *VIH2* expression in Arabidopsis using pVIH1::GUS and pVIH2::GUS reporter lines. We found *VIH2* to be expressed in different tissues and organs, and strong expression for *VIH1* in pollen as previously reported (*Laha et al., 2016*) (*Figure 1—figure supplement 3*). However, our GUS lines reveal additional expression of *VIH1* in root tips, where also *VIH2* is expressed (*Figure 1—figure supplement 3*). VIH1 and VIH2 reporter lines harboring a C-terminal mCitrine (mCit) tag (which complements the *vih1 vih2* mutant phenotype, compare *Figure 2A*, see below) reveal that VIH1 is expressed in pollen grains, pollen tubes and in the root tip, where VIH2 is also present (*Figure 1—figure supplement 4*). Thus, VIH1 and VIH2 show partially overlapping expression in different tissues and organs and both enzymes are localized in the cytoplasm (*Figure 1—figure supplement 4*; compare Figure 4A). This prompted us to generate *vih1 vih2* double mutants: We characterized *vih1-2*, *vih1-6* and *vih2-4* T-DNA insertion lines as knock-out mutants of the respective enzyme (*Figure 1A,B*).

We found that *vih1-2 vih2-4* and *vih1-6 vih2-4* double mutants displayed severe growth phenotypes, while *vih1* and *vih2* single mutants and *vih1 vih2-6* double mutants looked similar to wild-type plants (*Figure 1C*). The VIH2 kinase domain is still expressed in the *vih2-6* mutant background, and thus *vih2-6* does not represent a full knock-out (see below, *Figure 2—figure supplement 1*). Next, we found that *vih2-4* plants expressing an inducible artificial microRNA (amiRNA) targeting the VIH1 transcript show intermediate growth phenotypes (*Figure 1—figure supplement 5*). At seedling stage, the *vih1-2 vih2-4* double mutant displayed a severe growth phenotype, while *vih1-2*, *vih2-4* and *ipk1-1* looked more similar to wild-type (*Figure 1D*). *ipk1-1* represents a mild loss-of-function mutant allele of Arabidopsis inositol-pentakisphosphate 2-kinase 1. Interestingly, while wild-type, *vih1-2*, *vih2-4* and *ipk1-1* seedlings showed a decrease in root length in response to Pi limitation, *vih1-2 vih2-4* showed the opposite trend (*Figure 1E*). Taken together, independent genetic

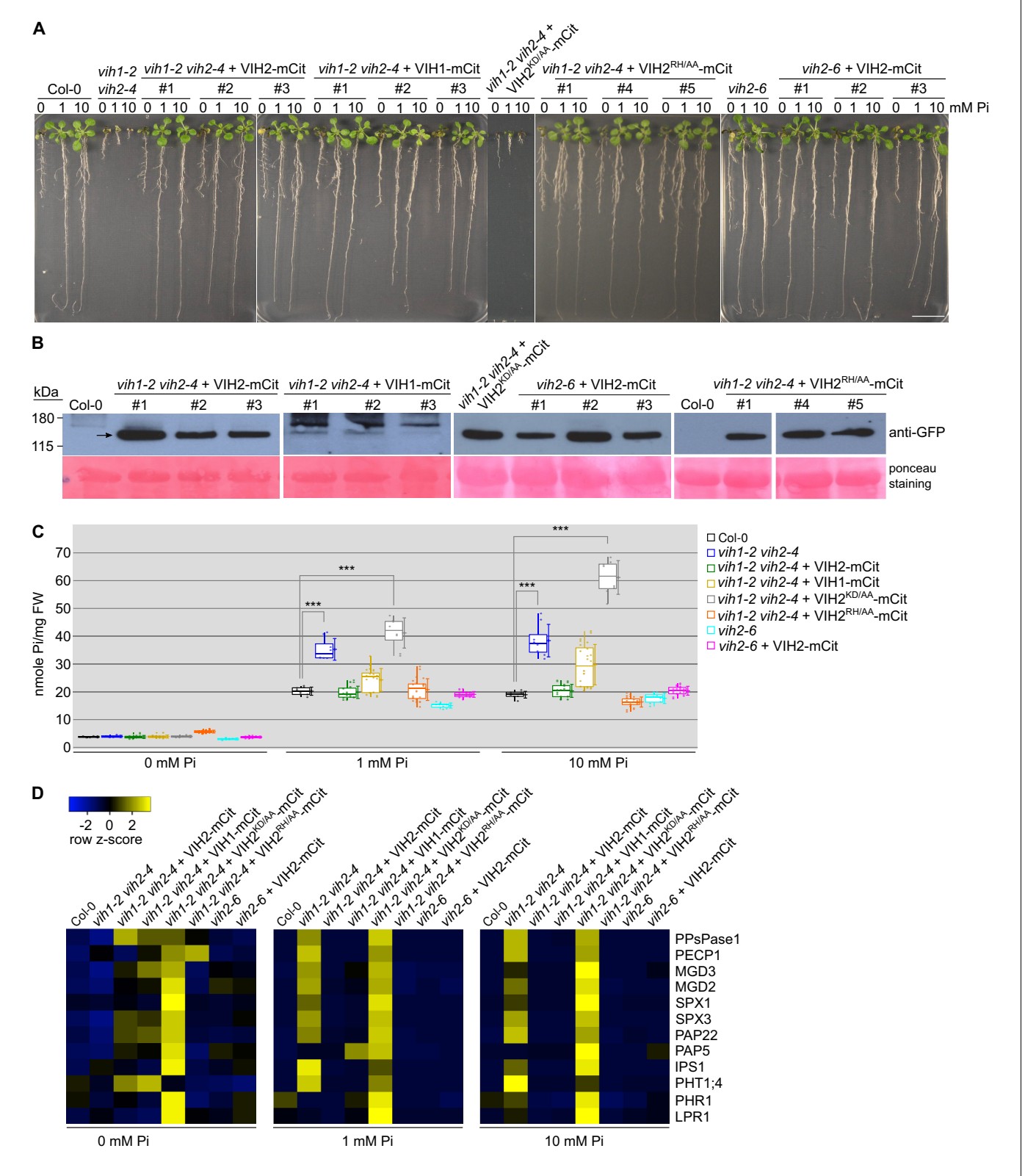

**Figure 2.** VIH kinase and phosphatase activities regulate plant Pi homeostasis. (**A**) Complementation of *vih1 vih2* growth phenotypes. Shown are three independent lines for each construct grown in different Pi regimes. Plants were germinated in vertical ¹/²MS plates for 8 d, transferred to Pi-free ¹/²MS plates supplemented with either 0 mM, 1 mM or 10 mM Pi and grown for additional 6 d. Scale bars correspond to 2 cm. (**B**) Western blot showing the expression of VIH2-mCit, VIH1-mCit, VIH2$^{KD/AA}$-mCit, VIH2$^{RH/AA}$-mCit proteins (indicated by arrow heads) in the transgenic lines from (**A**) using an anti-

*Figure 2 continued on next page*

*Figure 2 continued*

GFP antibody. A Ponceau stain of the membrane is shown as loading control below. (C) Pooled Pi content of seedlings 14 DAG as shown in (A). For each position, four independent plants from each trangenic line were measured with two technical replicates. (D) qRT-PCR quantification of the PSI marker genes in seedlings shown in (A). Expression levels are represented as *Z*-scores. The original qRT-PCR data are shown in ***Supplementary file 3b***.

DOI: https://doi.org/10.7554/eLife.43582.008

The following figure supplements are available for figure 2:

**Figure supplement 1.** The *vih2-6* transcript encodes a truncated VIH2 protein harboring the kinase domain only.

DOI: https://doi.org/10.7554/eLife.43582.009

**Figure supplement 2.** Growth phenotypes of soil-grown *vih1-2 vih2-4* double mutant lines complemented with pVIH1::VIH1-mCit, pVIH2::VIH2-mCit, pVIH2::VIH2$^{KD/AA}$-mCit or pVIH2::VIH2$^{RH/AA}$-mCit.

DOI: https://doi.org/10.7554/eLife.43582.010

**Figure supplement 3.** The kinase and phosphatase activities of VIH1 and VIH2 together control plant Pi homeostasis at seedling stage.

DOI: https://doi.org/10.7554/eLife.43582.011

**Figure supplement 4.** The kinase and phosphatase activities of VIH1 and VIH2 together control Pi homeostasis in soil-grown adult plants.

DOI: https://doi.org/10.7554/eLife.43582.012

experiments suggest that VIH1 and VIH2 act redundantly in Arabidopsis PP-InsP metabolism, and that simultaneous loss-of-function of these conserved enzymes strongly affects plant growth.

We next studied Pi-related phenotypes in our mutant backgrounds. In Pi-deficient medium supplemented with either 1 mM or 10 mM Pi, we found cellular Pi levels to be significantly increased in *vih1-2 vih2-4* and in *ipk1-1* seedlings (*Kuo et al., 2014*), but not in *vih1* or *vih2* single mutants, when compared to wild-type (*Figure 1F*). In the 10 mM Pi condition, the *vih1-2 vih2-4* double mutant accumulated significantly more Pi also when compared to *ipk1-1*. In a genetically independent experiment, estradiol-induced expression of an amiRNA targeting the *VIH1* transcript in the *vih2-4* single mutant background also led to a significant increase in shoot Pi content when compared to the non-induced control (*Figure 1—figure supplement 5*). We hypothesized that the increase in cellular Pi in *vih1-2 vih2-4* plants may be due to the double mutant lacking a VIH1 and VIH2-generated PP-InsP isoform critically involved in Pi homeostasis and starvation responses (*Wild et al., 2016*). Consistently, we found PSI gene expression misregulated in the *vih1-2 vih2-4* double mutant (*Figure 1G*). PSI marker genes were constitutively expressed in *vih1-2 vih2-4* seedlings grown in Pi sufficient (1 mM, 10 mM) growth conditions (*Figure 1G*), suggesting that the negative regulation of PHR1/PHL1 by PP-InsP-bound SPX domains is compromised (*Wild et al., 2016*; *Qi et al., 2017*). Together, these experiments implicate VIH1 and VIH2-generated PP-InsPs in Pi homeostasis and starvation responses.

## The VIH kinase and phosphatase domains together control plant Pi homeostasis

We next complemented *vih1-2 vih2-4* mutant lines with VIH1/VIH2 wild-type constructs containing a C-terminal mCit fluorescent tag (*Figure 2A*). At seedling stage, expression of wild-type VIH1 and VIH2 from their endogenous promoters rescued the growth phenotype, cellular Pi content and PSI gene expression of the double mutant (*Figure 2A–D*). We could not complement *vih1-2 vih2-4* with constructs expressing a VIH2$^{KD/AA}$ version containing point-mutations in the kinase domain (*Figure 2A–D*). In contrast, expression of a construct harboring point mutations in the VIH2 phosphatase domain (VIH2$^{RH/AA}$) complemented *vih1-2 vih2-4*-related phenotypes (*Figure 2A–D*). We next analyzed the *vih2-6* allele, also recently described as *vip1-1* (*Kuo et al., 2018*). We found that *vih2-6* results in a shortened transcript, which may encode a truncated version of VIH2 harboring the PPIK5K kinase domain only (*Figure 2—figure supplement 1*). No phenotypes significantly different from the Col-0 wild-type control plants were observed for the *vih2-6* allele at seedling stage (*Figure 2A–D*). However, when we re-performed our analyses on soil-grown adult plants, we found that our VIH2$^{RH/AA}$ complementation lines and the *vih2-6* mutant displayed a significantly reduced shoot Pi content when compared to wild-type (*Figure 2—figure supplement 2*). *vih2-6* phenotypes were rescued by expressing wild-type VIH2 in the *vih2-6* mutant background (*Figure 2—figure supplement 2*).

To further characterize the relative contributions of the VIH2 kinase and phosphatase domains to Pi homeostasis, we compared constitutively over-expressed wild-type VIH2, VIH2$^{KD/AA}$ and VIH2$^{RH/AA}$ in

Col-0 wild-type background. We found that VIH2$^{KD/AA}$ seedlings showed significantly increased cellular Pi levels, while VIH2$^{RH/AA}$ contained less Pi when grown in Pi sufficient conditions (*Figure 2—figure supplement 3*). Plants over-expressing wild-type VIH2 had Pi levels similar to Col-0 control seedlings (*Figure 2—figure supplement 3*). PSI target gene expression was found misregulated in both VIH2$^{KD/AA}$ and VIH2$^{RH/AA}$ over-expression lines (*Figure 2—figure supplement 3*). At adult stage, mutations in the kinase and phosphatase domains of VIH1 and VIH2 all lead to visible growth phenotypes, with plants having smaller rosette leaves (*Figure 2—figure supplement 4*). The shoot Pi content in VIH1$^{KD/AA}$ and VIH2$^{KD/AA}$ plants was significantly increased compared to wild-type (*Figure 2—figure supplement 4*). The Pi content of the VIH1$^{RH/AA}$ and VIH2$^{RH/AA}$ lines appeared however similar to the Col-0 control when plants were grown in soil (*Figure 2—figure supplement 4*). Taken together, our complementation and overexpression analyses reveal an essential role for the VIH kinase domain, while we speculate that the VIH phosphatase domain may have a regulatory function in plant Pi homeostasis and signaling.

## VIH1/VIH2 are part of the plant Pi starvation response pathway

To test our hypothesis that diphosphoinositol pentakisphosphate kinases are part of the plant Pi starvation response, we next analyzed the genetic interaction between VIH1, VIH2 and the known phosphate starvation response transcription factors PHR1 and PHL1 (*Bustos et al., 2010*). We found that at seedling stage *vih1-2 vih2-4 phr1 phl1* quadruple mutants partially rescue the seedling growth phenotype of *vih1-2 vih2-4* (*Figure 3A*). Cellular Pi levels are significantly increased in the *vih1-2 vih2-4* double mutant (see above) and significantly reduced in the *phr1 phl1* double mutant (*Bustos et al., 2010*) when compared to the Col-0 wild-type control (*Figure 3B*). The *vih1-2 vih2-4* root growth phenotype under different Pi regimes (compare *Figure 1E*, see above) is restored in the quadruple mutant (*Figure 3C*). As expected, the high PSI gene expression observed for *vih1-2 vih2-4* plants in Pi sufficient growth conditions is suppressed in the quadruple mutant (*Figure 3D*). A partial rescue of the *vih1-2 vih2-4* growth phenotype was observed for the quadruple mutant also at later developmental stages, with Pi shoot contents being significantly increased compared to the Col-0 wild-type control (*Figure 3—figure supplement 1*).

We next measured PP-InsP levels in Col-0 wild-type, *vih1-2 vih2-4* and *vih1-2 vih2-4 phr1 phl1* Arabidopsis plants, by feeding $^3$H labeled inositol to 7 DAG seedlings for an additional 5 d under Pi sufficient conditions (see Materials and methods). We observed higher amounts of InsP$_7$ isoforms in our double and quadruple mutant plants when compared to wild-type, while InsP$_8$ isoforms were no longer detectable (*Figure 3E*). Quantification of the InsP$_7$/InsP$_6$ ratios revealed significantly higher InsP$_7$ levels in the double and quadruple mutant (*Figure 3F*). Quantification of the InsP$_8$/InsP$_7$ ratios supports the notion that InsP$_8$ isoforms are not detectable in our *vih1-2 vih2-4* and *vih1-2 vih2-4 phr1 phl1* mutants (*Figure 3G*). Due to the low labeling efficiency, we could not investigate PP-InsP levels under Pi starvation.

Taken together, our genetic and metabolic analyses suggest that VIH1/VIH2 and PHR1/PHL1 are part of a common genetic pathway and that in our double and quadruple mutants the InsP$_8$ signaling molecule is not detectable.

## VIH1/VIH2 transcription and protein abundance are not affected by Pi levels

We next tested if VIH1 and 2 promoter activities and protein levels change in response to different Pi growth regimes. We generated pVIH2::H(istone)2b-mCit lines, which accumulate the fluorescence signal in the nucleus, in either wild-type or *vih2-4* mutant background and found no significant changes in promoter activity in low to high Pi growth conditions (*Figure 4—figure supplement 1*). In contrast, the known PSI targets SPX1 and SPX3 are highly induced under Pi starvation (*Figure 4—figure supplement 1*) (*Duan et al., 2008*). In agreement with these findings, our GUS reporter lines show slightly reduced VIH1 and VIH2 promoter activities in Pi starvation conditions (*Figure 4—figure supplement 1*). VIH1-mCit and VIH2-mCit fusion proteins under the control of their native promoters are expressed to similar levels in plants grown in different Pi regimes (*Figure 4A*), while SPX1 and SPX3 accumulate under Pi starvation (*Figure 4B*). We next introduced pSPX1::SPX1-eGFP and pSPX3::SPX3-eGFP into our *vih1-2 vih2-4* double mutant, and found the PSI marker proteins to be highly abundant in the double mutant background when expressed under Pi repleted conditions

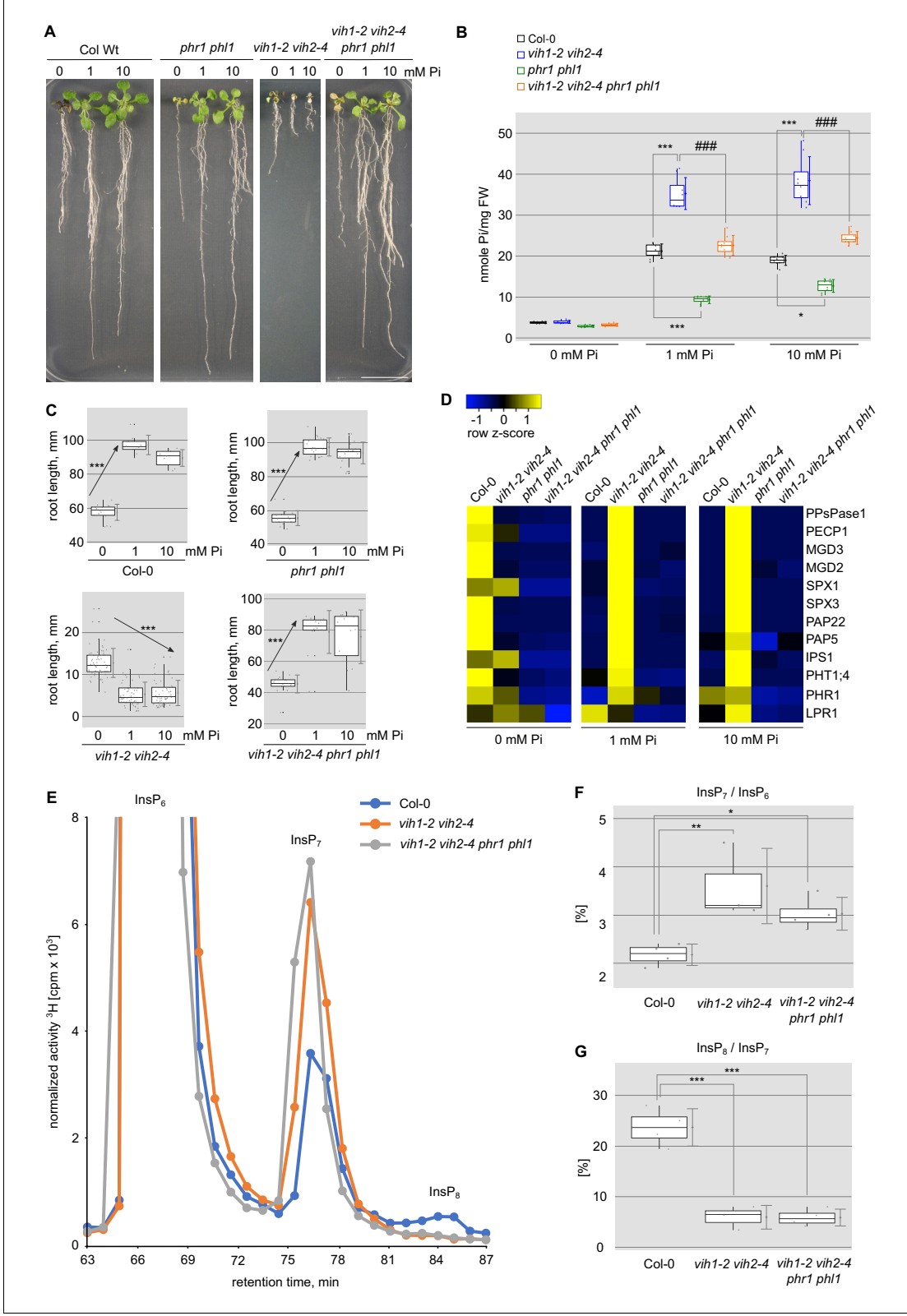

**Figure 3.** Deletion of PHR1 and PHL1 rescues *vih1-2 vih2-4* growth phenotypes. (**A**) Growth phenotype of Col-0, *phr1 phl1*, *vih1-2 vih2-4*, and *vih1-2 vih2-4 phr1 phl1* seedlings grown in different Pi conditions. Seedlings 7 DAG were transplanted from ¹/²MS plates to Pi-deficient ¹/²MS liquid supplemented with 0 mM, 1 mM or 10 mM Pi, and grown for additional 6 d. (**B**) Cellular Pi content of seedlings 14 DAG described in (**A**). For each Pi concentration, at least four independent plants were measured with two technical replicates each. (**C**) Root length of seedlings 14 DAG described in

*Figure 3 continued on next page*

*Figure 3 continued*

(A). For each Pi concentration, seedlings from at least three independent plates were analyzed. (D) qRT-PCR quantification of the PSI marker genes in seedlings shown in (A). Expression levels are represented as Z-scores. The original data of qRT-PCR are shown in **Supplementary file 3c**. (E) Normalized HPLC profiles of [$^3$H] inositol-labeled Col-0 (blue line), *vih1-2 vih2-4* double mutant (orange line) and *vih1-2 vih2-4 phr1 phl1* quadruple mutant (gray line). The InsP$_6$, InsP$_7$ and InsP$_8$ regions are shown (termed as such, as specific PP-InsP$_5$/(PP)$_2$-InsP$_4$ regioisomers cannot be resolved using this method). Fractions were collected each minute (solid dots) for radioactivity determination. The experiment was repeated at least three times with similar results, and representative profiles from one experiment are shown. The InsP$_7$ / InsP$_6$ ratio is plotted in (F) and the InsP$_8$ / InsP$_7$ ratio in (G).
DOI: https://doi.org/10.7554/eLife.43582.017

The following figure supplement is available for figure 3:

**Figure supplement 1.** Growth phenotypes of soil-grown *phr1 phl1*, *vih1-2 vih2-4* and *vih1-2 vih2-4 phr1 phl1* mutant adult plants.
DOI: https://doi.org/10.7554/eLife.43582.018

(*Figure 4—figure supplement 2*). A pPHR1::eGFP-PHR1 control line did not show this behavior (*Figure 4—figure supplement 2*).

We next speculated that similar to what has been reported for human PPIP5Ks, plant VIH1/VIH2 kinase and phosphatase activities may be regulated at the enzyme kinetic level (*Gu et al., 2017a*; *Shears, 2018*). To investigate if the regulation of plant Pi homeostasis may be similar to the human system, we complemented our *vih1-2 vih2-4* mutant with HsPPIP5K2, codon-optimized for Arabidopsis, and expressed under the control of the Arabidopsis VIH2 promoter (*Figure 4C*; *Figure 4—figure supplement 3*). We found that expression of wild-type HsPPIP5K2 or HsPPIP5K2$^{KD/AA}$ carrying point-mutations that target the kinase domain, did not rescue the *vih1-2 vih2-4* mutant phenotype (*Figure 4—figure supplement 3*). Notably, expression of a HsPPIP5K2$^{RH/AA}$ mutant in the phosphatase domain partially complemented the *vih1-2 vih2-4* growth phenotype (*Figure 4C*), and restored cellular Pi levels (*Figure 4D*). This indicates that human and plant PPIP5Ks may generate similar PP-InsP signaling molecules, able to regulate Pi homeostasis in these different organisms (*Gu et al., 2017a*). Together, these experiments indicate that VIH enzyme activity is not simply controlled at the transcript or protein level, but most likely through the enzymatic activity of their kinase and phosphatase domains.

## Arabidopsis VIH2 is a 1PP-InsP kinase

The enzymatic properties of Arabidopsis VIH1 and VIH2 have not been characterized thus far. We mapped the catalytic PPIP5K kinase and phosphatase domains using the program HHPRED (*Zimmermann et al., 2018*) and expressed the isolated domains in insect cells (*Figure 5—figure supplement 1*, see Materials and methods). Next, we determined the substrate specificity of the VIH2 kinase domain (VIH2-KD, residues 11–338, *Figure 5—figure supplement 1A,C*) in quantitative nuclear magnetic resonance (NMR) assays using $^{13}$C labeled InsP substrates (see Materials and methods). In our assays, we found that AtVIH2-KD catalyzes the synthesis of 1PP-InsP$_5$ from InsP$_6$ and InsP$_8$ from 5PP-InsP$_5$ (*Figure 5A,B*; *Figure 5—figure supplement 2*). We recorded similar activities for human PPIP5K2 (*Figure 5A,B*; *Figure 5—figure supplement 1*), in agreement with earlier reports (*Choi et al., 2007*; *Fridy et al., 2007*; *Lin et al., 2009*; *Wang et al., 2012*). The AtVIH2-KD$^{KD/AA}$ mutant protein (*Figure 5—figure supplement 1*) shows no detectable activity using InsP$_6$ as a substrate, and a minuscule activity towards 5PP-InsP$_5$ (*Figure 5—figure supplement 2*). NMR-derived enzyme kinetics suggest 5PP-InsP$_5$ to represent a better substrate for AtVIH2 when compared to InsP$_6$ (*Figure 5C*, *Figure 5—figure supplement 2*). Together with the in planta observation that InsP$_8$ levels are reduced and InsP$_7$ levels are increased in *vih1-2 vih2-4* mutant seedling (*Figure 3E–G*), our data reveal that the Arabidopsis VIH kinase domains generate InsP$_8$ .

## The VIH2 phosphatase domain has 1 and 5PP-InsPase activity

PPIP5Ks from different organisms all contain a C-terminal phosphatase domain (*Pascual-Ortiz et al., 2018*; *Shears et al., 2017*). Inositol pyrophosphatase activity has been established for fission yeast Asp1 and for human PPIP5K1 (*Wang et al., 2015*; *Pascual-Ortiz et al., 2018*). The VIH2 phosphatase domain recombinantly expressed and purified from insect cells (VIH2-PD, residues 359–1002, *Figure 5—figure supplement 1*) hydrolyzes both 1PP-InsP$_5$ and 5PP-InsP$_5$, yielding InsP$_6$, but not the non-hydrolyzable 5PCP-InsP$_5$ analog (*Figure 5D,E*) (*Wu et al., 2016*). The *Saccharomyces cerevisiae* Vip1 phosphatase domain (ScVip1-PD, residues 515–1088) (*Figure 5—figure supplement 1*),

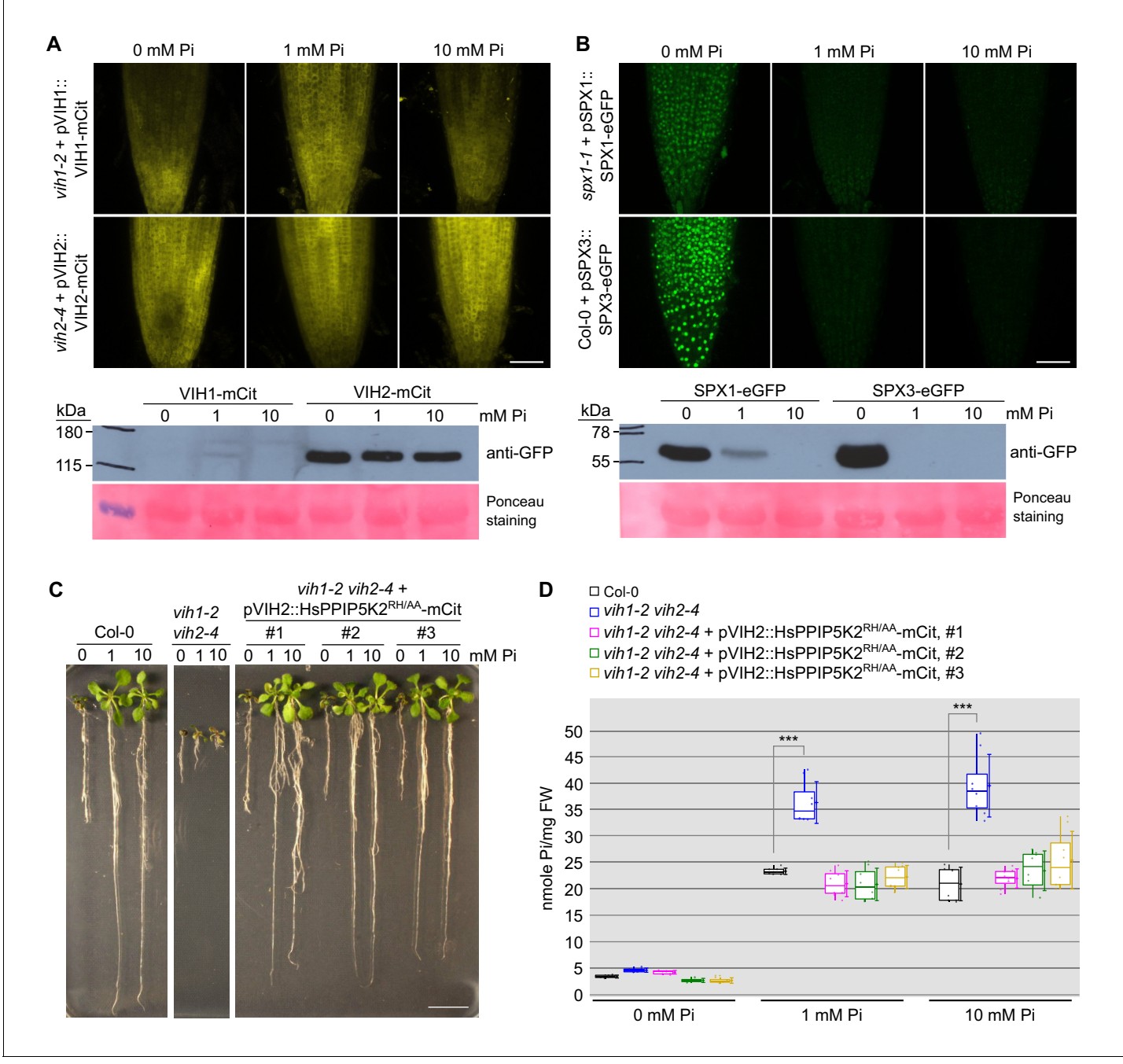

**Figure 4.** VIH1/VIH2 protein levels remain stable in different Pi growth regimes. (**A**) Representative confocal scanning microscopy images showing a mCit fluorescent signal in the root tip of *vih1-2* and *vih2-4* seedlings transformed with pVIH1::VIH1-mCit or pVIH2::VIH2-mCit respectively. Seedlings 7 DAG were transferred from $^{1/2}$MS plates to Pi-deficient $^{1/2}$MS liquid medium supplemented with 0 mM, 1 mM or 10 mM Pi, and grown for 3 d. Scale bar = 50 μm. Protein levels of VIH1-mCit and VIH2-mCit in plants detected by an anti-GFP antibody are shown below, the ponceau stained membrane serves as loading control. (**B**) Representative confocal scanning microscopy images showing a GFP fluorescent signal in the root tip of *spx1-1* and Col-0 wild-type seedlings transformed with pSPX1::SPX1-eGFP and pSPX3::SPX3-eGFP respectively. Seedlings were treated as described in (**E**). Scale bar = 50 μm. Protein levels of SPX1-eGFP and SPX3-eGFP in plants detected by an anti-GFP antibody are shown below, the ponceau stained membrane is included as loading control. (**C**) Growth phenotypes of Col-0, *vih1-2 vih2-4* and 3 independent lines of *vih1-2 vih2-4* seedlings complemented with pVIH2::HsPPIP5K2^RH/AA^-mCit. Seedlings 7 DAG were transplanted from $^{1/2}$MS plates to Pi-deficient $^{1/2}$MS liquid supplemented with 0 mM, 1 mM or 10 mM Pi, and grown for additional 6 d. Scale bars correspond to 2 cm. (**D**) Pi content of seedlings 14 DAG as described in (**D**). Four independent plants from each transgenic line were measured with two technical replicates each.

DOI: https://doi.org/10.7554/eLife.43582.013

*Figure 4 continued on next page*

*Figure 4 continued*

The following figure supplements are available for figure 4:

**Figure supplement 1.** VIH1 and VIH2 promoter activities are not induced in low or high Pi conditions.

DOI: https://doi.org/10.7554/eLife.43582.014

**Figure supplement 2.** PSI marker proteins SPX1 and SPX3 accumulate in the *vih1-2 vih2-4* mutant seedlings grown in Pi replete condition.

DOI: https://doi.org/10.7554/eLife.43582.015

**Figure supplement 3.** Growth phenotypes of soil-grown *vih1-2 vih2-4* double mutant lines complemented with pVIH2::HsPPIP5K2-mCit, pVIH2::HsPP5PIK2$^{KD/AA}$-mCit or pVIH2::HsPPIP5K2$^{RH/AA}$-mCit.

DOI: https://doi.org/10.7554/eLife.43582.016

which shares only 24% sequence identity with the plant enzyme (*Figure 1—figure supplement 1*), showed a very similar substrate specificity in gel-based assays (*Figure 5D,E*). It is of note that specific 1-phosphatase activity has been previously reported for the isolated Asp1 phosphatase domain from *Schizosaccharomyces pombe* (*Wang et al., 2015*). Addition of either magnesium ions or a metal chelator had no detectable effect on VIH2 phosphatase activity during overnight incubation (*Figure 5E*). We next assayed if the VIH2-PD$^{RH/AA}$ mutations in the phosphatase domain (*Figure 5—figure supplement 1*) would abolish the phosphatase activity of the mutant proteins. We found both VIH2-PD$^{RH/AA}$ and ScVip1-PD$^{RH/AA}$ to still possess catalytic activity towards the 5PP-InsP$_7$ substrate (*Figure 5F*, *Figure 5—figure supplement 3*). Together, our in vitro assays of the isolated phosphatase domains suggest that the C-terminus of the bifunctional plant and yeast PPIP5K enzymes can hydrolyze different PP-InsP isoforms, but not InsP$_6$. In addition, the VIH2$^{RH/AA}$ mutant may still possess residual phosphatase activity in vivo.

## PPIP5Ks can respond to changes in cellular ATP and Pi levels

Finally, we wanted to investigate whether PPIP5K kinase and phosphatase domains cooperate in the context of the full-length enzyme. We could express full-length Arabidopsis VIH1 and 2 but found the purified protein to be aggregated in size-exclusion chromatography experiments (*Figure 6—figure supplement 1*). We thus produced the related full-length Vip1 from *S. cerevisiae*, which we could purify to homogeneity and which behaves as a monomer in size-exclusion chromatography (*Figure 6—figure supplement 1*). We next incubated ScVip1 with 5PP-InsP$_5$ while varying ATP substrate concentrations in gel-based activity assays. We found that at low ATP concentrations the ScVip1 phosphatase activity predominates, releasing InsP$_6$ (*Figure 6A*). Increasing ATP levels resulted in stimulation of the PP-InsP kinase activity, producing increasing amounts of InsP$_8$ (*Figure 6A*). Interestingly, at intermediate ATP concentrations, the kinase and phosphatase activities appear balanced, as we did not observe net production of either InsP$_6$ or InsP$_8$ (*Figure 6A*). To further substantiate these qualitative experiments, we next quantified conversion of 5PP-InsP$_5$ to InsP$_6$ in NMR experiments (see Materials and methods). We found low ATP-Mg$^{2+}$ concentrations to correlate with high ScVIP1 phosphatase activity, and *vice versa* (*Figure 6B*). Our results suggest that PPIP5K enzymes may relay changes in cellular ATP concentration to changes in PP-InsP levels, as previously reported for the human enzyme (*Gu et al., 2017a*). To test if this mechanism may be of relevance to plant Pi homeostasis, we next quantified the ATP/ADP levels in seedlings grown in Pi deficient medium, or after re-feeding with 0.5 mM Pi or 0.5 mM Phi, respectively (*Figure 6C*). We found, that ATP/ADP ratios increased after Pi re-feeding over a 48 hr time course, and were significantly higher when compared to the 0 mM Pi control (*Figure 6D*). Importantly, this was not the case when we used 0.5 mM Phi instead of Pi in the experiment (*Figure 6D*). We confirmed that the conditions chosen for this experiment were physiological relevant and found that upon Pi re-feeding plants showed significantly increased fresh weight, root length and cellular Pi content (*Figure 6F*). The Pi content was significantly increased compared to the 0 mM Pi control already at 8 hr after re-feeding, while significant differences in seedling fresh weight and root length became apparent after 48 hr (*Figure 6C,F*). No such effect was observed when using 0.5 mM Phi instead of Pi (*Figure 6C, F*), despite both molecules suppressing PSI gene expression (*Figure 6E*).

To rationalize the puzzling finding that Phi and Pi can both suppress Pi starvation responses, yet only Pi can promote physiological changes, we next tested if Pi and/or Phi would have an effect on PPIP5K enzyme activity. We found that 5PP-InsP$_5$ hydrolysis by the isolated ScVip1 phosphatase domain is inhibited in the presence of 10 mM Pi in quantitative NMR time course experiments

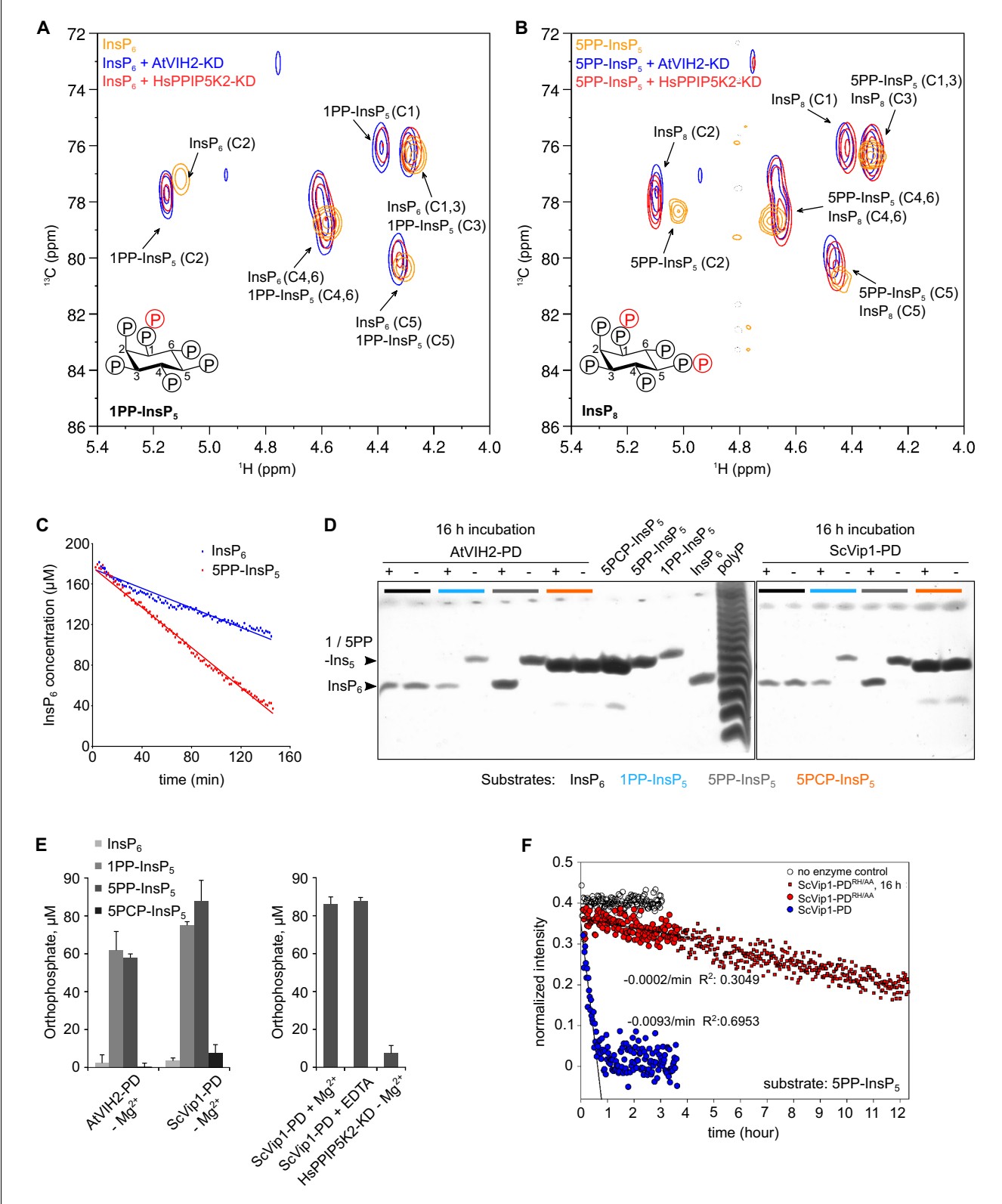

**Figure 5.** The AtVIH2 kinase domain has 1-kinase activity and produces 1PP-InsP$_5$ and InsP$_8$, the phosphatase domain is a 1 - and 5 - pyrophosphatase. (**A, B**) 2D $^1$H-$^{13}$C-HMBC spectra of the products produced by plant AtVIH2-KD (blue trace) and human HsPPIP5K2-KD (red trace) in the presence of InsP$_6$ (**A**) or 5PP-InsP$_5$ (**B**). Substrate standards are colored in yellow. (**C**) Decay of the InsP$_6$ or 5-PP-InsP$_5$ substrate during the NMR-time course experiment shown in (**A**) and (**B**), respectively. A fit of the initial decay indicates a turnover number of ~0.4/min with InsP$_6$ as a substrate and ~1/min

*Figure 5 continued on next page*

*Figure 5 continued*

using 5-PP-InsP$_5$ as a substrate. (**D**) Qualitative native PAGE phosphatase activity assay. Reactions containing recombinant phosphatase domain of AtVIH2 (AtVIH2-PD,~5 μg) or ScVip1 (ScVip1-PD,~27 μg) were incubated with 175 μM of either InsP$_6$ (black), 1PP-InsP$_5$ (blue), 5PP-InsP$_5$ (gray) or a non-hydrolyzable 5PCP-InsP$_5$ analog (orange) for 16 hr at 37˚C. 40 μl of the reaction were separated in a 35% acrylamide gel. The bands corresponding to InsP$_6$ and 1 or 1/5PP-InsP$_5$ are indicated by an arrowhead. (**E**) Malachite green-based phosphatase activity assay. Reactions containing recombinant AtVIH2-PD (~5 μg) or ScVip1-PD (~27 μg) were incubated with 175 μM InsP$_6$, 1PP-InsP$_5$, 5PP-InsP$_5$ or 5PCP-InsP$_5$ for 16 hr at 37˚C (left). 1 mM Mg$^{2+}$ or 5 mM EDTA were supplemented as indicated (right; 5PP-InsP$_5$ only). Recombinant HsPPIP5K2-KD (~17 μg) was used as a negative control and tested only with 5PP-InsP$_5$ (right). Reactions were performed in quadruplicates and released orthophosphate was quantified using a malachite green assay (*Baykov et al., 1988*). (**F**) NMR time course experiment comparing the phosphatase activities of 2 μM ScVip1-PD and ScVip1-PD$^{RH/AA}$ using 40 μM [$^{13}$C$_6$]5PP-InsP$_5$ as substrate. Samples were measured in a pseudo-2D spin-echo difference experiment and the relative intensities of the C2 peaks of InsP$_6$ and 5PP-InsP$_5$ were quantified.

DOI: https://doi.org/10.7554/eLife.43582.019

The following figure supplements are available for figure 5:

**Figure supplement 1.** The purification of recombinant proteins.

DOI: https://doi.org/10.7554/eLife.43582.020

**Figure supplement 2.** 1D $^{31}$P and 2D $^1$H-$^{31}$P-HMBC spectra of the products produced by AtVIH2-KD, and 2D $^1$H-$^{13}$C-HMBC spectra of the products produced by plant AtVIH2-KD$^{KD/AA}$.

DOI: https://doi.org/10.7554/eLife.43582.021

**Figure supplement 3.** The ScVip1-PD$^{RH/AA}$ and AtVIH2-PD$^{RH/AA}$ recombinant enzymes show reduced phosphatase activity.

DOI: https://doi.org/10.7554/eLife.43582.022

(*Figure 6—figure supplement 2*). This prompted us to analyze the enzymatic reaction profiles of full-length ScVip1 in the presence of 0 and 10 mM Pi, or in the presence of 10 mM Phi. We found that addition of Pi or Phi promoted the synthesis of InsP$_8$, which based on our in vivo and in vitro analyses represents the active signaling molecule in plant Pi homeostasis and starvation responses (*Figure 6—figure supplement 2*). Thus, Phi may suppress PSI gene expression by stimulating the kinase activities of VIH1 and VIH2, even though the cellular ATP/ADP ratio remains relatively constant (*Figure 6D*). In physiologically relevant growth conditions, Pi and ATP/ADP levels may together shape the output of the Pi starvation response, by controlling the synthesis and breakdown of the InsP$_8$ signaling molecule catalyzed by VIH1 and VIH2.

## Discussion

Plants evolved multiple strategies to maintain cellular Pi homeostasis in Pi deficient conditions, controlled by PHR family transcription factors, their SPX domain interaction partners, phosphate transporters, ubiquitin ligases controlling phosphate transporter protein levels, microRNAs and ferroxidase enzymes (*Puga et al., 2017*). The recent finding that eukaryotic SPX domains are cellular receptors for PP-InsPs (*Wild et al., 2016*) and that PP-InsP binding can regulate the activity of PHR transcription factors (*Puga et al., 2014*; *Wang et al., 2014*; *Wild et al., 2016*; *Qi et al., 2017*) and phosphate transporters (*Wild et al., 2016*; *Potapenko et al., 2018*) prompted us to investigate if the Arabidopsis diphosphoinositol pentakisphosphate kinases VIH1 and 2 are components of the plant phosphate starvation response pathway. Our experiments demonstrate that VIH1 and VIH2 redundantly regulate Pi homeostasis, growth and development.

Deletion of the phosphate starvation response transcription factors PHR1 and PHL1 (*Rubio et al., 2001*; *Bustos et al., 2010*) can partially rescue the *vih1-2 vih2-4* mutant phenotype. This confirms that VIHs, their PP-InsP reaction products and PHR1/PHL1 are part of a common signaling cascade. Our observation that *vih1-2 vih2-4* mutant seedlings show constitutive Pi starvation responses (*Figure 1G*), and less severe root phenotypes in Pi deficient conditions (*Figure 1E*), suggests that *vih1-2 vih2-4* seedling may hyper-accumulate Pi to toxic levels. It is of note that deletion of inositol hexakisphosphate kinases 1 and 2 from human cells, which blocks synthesis of InsP$_7$ and InsP$_8$ isoforms, also lead to elevated cellular Pi levels (*Wilson et al., 2019*). Our work is in good agreement with the recent observation that mutations in the inositol polyphosphate biosynthesis pathway impact Pi signaling (*Kuo et al., 2018*). The fact that our quadruple mutant does not resemble wild-type plants may indicate that PP-InsPs regulate the activity of other components of the Pi starvation response, such as additional transcription factors (*Sun et al., 2016*), Pi transporters (*Hamburger et al., 2002*; *Liu et al., 2015*; *Liu et al., 2016*; *Wild et al., 2016*), or other SPX-domain

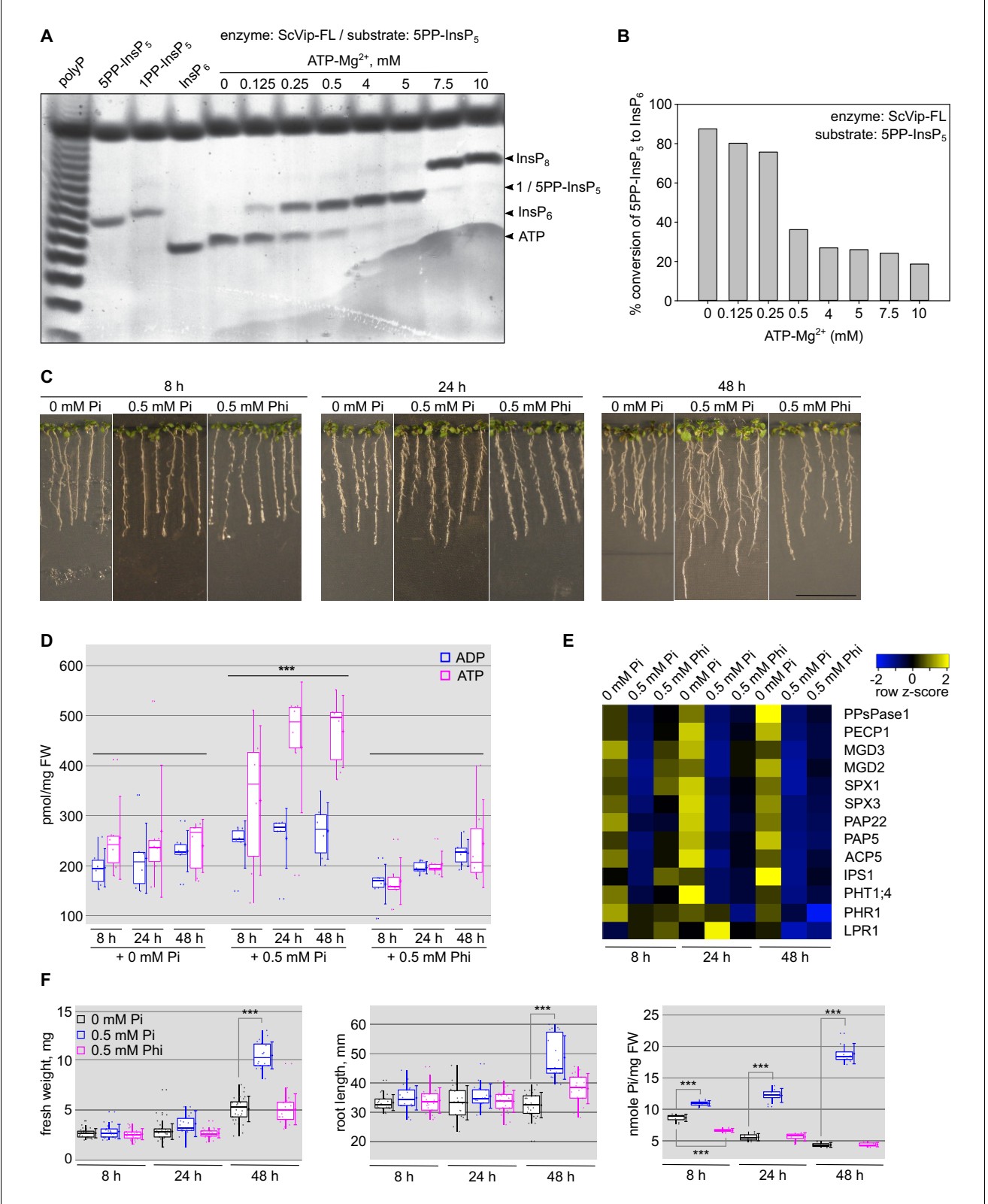

**Figure 6.** Changes in cellular ATP levels affect the relative PPIP5K kinase and phosphatase activities. (**A**) Bi-functional PP-InsP activity assay of ScVip1. Reactions containing 2 μM protein, 40 μM 5PP-InsP$_5$ and various ATP-Mg$^{2+}$ concentrations were incubated at 37°C for 45 min. Product PP-InsPs were separated on a native PAGE gel and stained with toluidine blue. The bands corresponding to InsP$_6$, 1/5PP-InsP$_5$, InsP$_8$ and ATP are indicated by arrow heads. (**B**) Quantification of conversion of the 5-PP-InsP$_5$ (substrate) to InsP$_6$ (product) by full-length ScVip1 (FL) enzyme in reactions containing different

*Figure 6 continued on next page*

*Figure 6 continued*

concentrations of ATP-Mg$^{2+}$, recorded by NMR specroscopy. (**C**) Col-0 seedlings 6 DAG were transplanted from $^{1/2}$MS plates to Pi-deficient $^{1/2}$MS plates and incubate 3 days more. Then, the seedlings were transplanted again to Pi-deficient $^{1/2}$MS plates supplemented with 0 mM Pi, 0.5 mM Pi or 0.5 mM Phi, and grown for additional 8 hr, 24 hr or 48 hr. (**D**) Determination of the cellular ATP and ADP concentrations of the seedlings shown in (**C**). (**E**) qRT-PCR quantification of PSI marker genes in the seedlings shown in (**C**). Expression levels are represented as Z-scores. The original qRT-PCR data are shown in **Supplementary file 3e**. (**F**) Quantification of seedling fresh weight, primary root length and cellular Pi concentrations of the plants shown in (**C**).

DOI: https://doi.org/10.7554/eLife.43582.023

The following figure supplements are available for figure 6:

**Figure supplement 1.** Purification of full-length AtVIH1, AtVIH2 and ScVip1 proteins from insect cells.

DOI: https://doi.org/10.7554/eLife.43582.024

**Figure supplement 2.** Pi inhibits the ScVip1-PD phosphatase activity, but promotes synthesis of InsP$_8$ catalyzed by full-length ScVip1.

DOI: https://doi.org/10.7554/eLife.43582.025

containing proteins (*Park et al., 2014*). The strong growth and developmental phenotypes of the *vih1-2 vih2-4* double mutant may also hint at additional signaling functions for PP-InsPs unrelated to Pi homeostasis (*Tan et al., 2007*; *Sheard et al., 2010*; *Mosblech et al., 2011*; *Laha et al., 2015*; *Laha et al., 2016*; *Wild et al., 2016*). Together, our genetic analysis firmly places VIH1 and VIH2 in the plant Pi starvation response pathway.

It is known that several PP-InsP isoforms are present in eukaryotic cells. We found increased InsP$_7$ and non-detectable InsP$_8$ levels in our *vih1-2 vih2-4* mutant in vivo. In vitro, the AtVIH2 kinase domain prefers 5PP-InsP$_5$ as substrate to generate InsP$_8$ (1,5(PP)$_2$-InsP$_4$). Importantly, the 1PP-InsP$_5$ kinase activity of VIHs is essential for plant growth and development, as mutations targeting the kinase active site cannot rescue the *vih1-2 vih2-4* mutant phenotypes. These findings are in agreement with previous reports showing that InsP$_7$ isoforms accumulate and InsP$_8$ levels decrease in the *vih2-4* single mutant (*Laha et al., 2015*). Together, these findings support a role for InsP$_8$ as a central signaling molecule in plant Pi sensing, homeostasis and starvation responses.

Eukaryotic diphosphoinositol pentakisphosphate kinases are bifunctional enzymes. The biochemical activities of plant, yeast and human PPIP5Ks kinase domains are indeed similar (*Figure 5A,B*) (*Mulugu et al., 2007*; *Pascual-Ortiz et al., 2018*; *Lin et al., 2009*). This notion is further supported by our observation that the *vih1-2 vih2-4* mutant can be complemented with human PPIP5K2 harboring an active kinase domain (*Figure 4C*). In addition to their 1PP-InsP$_5$ kinase domain, PPIP5Ks also harbor a C-terminal phosphatase domain (*Shears et al., 2017*). For fission yeast Asp1 and human PPIP5K2 it has been previously shown that the phosphatase domain has inositol pyrophosphatase activity, with a preference for C1 pyrophosphorylated substrates (*Wang et al., 2015*; *Gu et al., 2017a*; *Pascual-Ortiz et al., 2018*). In our biochemical assays using Arabidopsis VIH2 and yeast Vip1 phosphatase domains, we observed inositol pyrophosphatase activity against both C1 and C5 pyrophosphorylated substrates (*Figure 5D,E*). This suggests that the C-terminal phosphatase domain of diphosphoinositol pentakisphosphate kinases is able to dephosphorylate the reaction product of the kinase domain.

Previously, single point mutations targeting the phosphatase domain have been assayed in vivo and in vitro, with conflicting results (*Mulugu et al., 2007*; *Choi et al., 2007*; *Wang et al., 2015*; *Gu et al., 2017a*; *Pascual-Ortiz et al., 2018*). In our qualitative and quantitative phosphatase assays, we found the corresponding mutations in AtVIH2-PD$^{RH/AA}$ and ScVip1-PD$^{RH/AA}$ to still harbor residual enzymatic activity (*Figure 5F*, *Figure 5—figure supplement 3*). The catalytic mechanism of histidine acid phosphatase domains in diphosphoinositol pentakisphosphate kinases is currently unknown. We thus cannot rule out that the phosphatase mutant proteins expressed in our *vih1-2 vih2-4* complementation lines are still partially functional. In any case, we do observe statistically significant differences in Pi shoot content only in VIH2$^{RH/AA}$ complementation lines at adult stage, with young seedling behaving similar to wild-type (*Figure 2*, *Figure 2—figure supplement 2*). In line with this, our *vih2-6* mutant (*Kuo et al., 2018*), which may express a kinase-only VIH2 protein, shows a similar behavior (*Figure 2*, *Figure 2—figure supplement 2*). It is thus presently difficult to clearly assign a physiological function for the VIH1 and VIH2 phosphatase domains. However, the strict conservation of the bifunctionality in PPIP5Ks and the fact that we could rescue our *vih1-2 vih2-4* double mutant only with a phosphatase-mutated version of human PPIP5K2 (*Figure 4—figure supplement*

*3*) together suggest a potential regulatory role for the phosphatase domain, as previously characterized for the human and yeast enzymes (*Gu et al., 2017a*; *Pascual-Ortiz et al., 2018*).

It has been previously proposed that plants may directly sense cellular Pi levels by inorganic phosphate directly binding to SPX sensor domains (*Puga et al., 2014*; *Wang et al., 2014*). Our structural and quantitative biochemical characterization of SPX domains as receptors for PP-InsPs together with the new evidence presented here now argues for a function of PP-InsPs as central signaling molecules in plant Pi homeostasis and starvation responses. We speculate that in our *vih1-2 vih2-4* mutant PHR1/PHL1 are not under the negative regulation of SPX proteins, due to the lack of the InsP$_8$ signaling molecule normally promoting their interaction (*Puga et al., 2017*; *Wang et al., 2014*; *Wild et al., 2016*; *Qi et al., 2017*). In line with this, PSI gene expression is constitutively upregulated in the double mutant, and deletion of PHR1 and PHL1 can partially rescue the *vih1-2 vih2-4* mutant phenotypes (*Figure 3*). It has been well established in yeast that PP-InsP levels change in response to changes in Pi availability (*Lonetti et al., 2011*; *Wild et al., 2016*). How changes in Pi levels in planta may relate to accumulation or depletion of PP-InsPs is poorly understood (*Azevedo and Saiardi, 2017*), but our genetic findings now allow us to speculate that the VIH kinase and phosphatase activities may be involved in this process. In line with this, over-expression of kinase and phosphatase mutated versions of VIH2 lead to elevated and reduced cellular Pi levels, respectively (*Figure 2—figure supplement 3*). VIH1 and 2 protein levels are stable in different Pi growth conditions (*Figure 4*). Thus, concerted regulation of the enzymatic activities of the VIH PP-InsP kinase and phosphatase activities may antagonistically control Pi homeostasis, growth and development.

It is of note that cellular ATP levels are reduced when extracellular Pi becomes limiting (*Choi et al., 2017*; *Gu et al., 2017a*). Human cells deprived of PP-InsPs conversely showed increased amounts of ATP and elevated Pi levels (*Wilson et al., 2019*). We found cellular ATP levels to be lower in Arabidopsis seedling grown under Pi deficient when compared to Pi replete conditions (*Figure 6D*), in good agreement with what has been reported in other plant species, and in other organisms (*Duff et al., 1989*; *Choi et al., 2017*; *Gu et al., 2017a*).

This prompted us to assay the kinase and phosphatase activities of full-length Vip1 from yeast, which we can purify to homogeneity from insect cells, in the presence of different ATP concentrations. Indeed, we found that at low ATP-Mg$^{2+}$ levels, Vip1 mainly acts as PP-InsP phosphatase, while at higher ATP-Mg$^{2+}$ concentrations significant InsP$_8$ synthesis could be observed (*Figure 6A,B*). We thus speculate that eukaryotic diphosphoinositol pentakisphosphate kinases/phosphatases have evolved to precisely sense cytosolic ATP concentration, or ATP/ADP ratios, relating changes in Pi levels to changes in PP-InsP pools.

PPIP5Ks may also respond to changing Pi levels directly, as recently reported for human PPIP5K2 (*Gu et al., 2017a*). Using the isolated phosphatase and full-length ScVip1, we further characterized Pi as an inhibitor of the phosphatase activity, while promoting InsP$_8$ synthesis in the context of the full-length enzyme (*Figure 6—figure supplement 2*). This points to a complex regulation of PPIP5Ks in vivo, which merits further exploration in the future. Importantly, while transfer of seedlings to Phi containing media does not induce ATP level changes, or physiological responses (*Figure 6D,F*), the Phi re-feeding suppresses PSI starvation responses and PSI target gene expression (*Figure 6E*), as previously shown (*Ticconi et al., 2001*; *Jost et al., 2015*). Indeed, we found Phi to stimulate InsP$_8$ synthesis by full-length ScVip1 to a similar extend as the Pi control (*Figure 6—figure supplement 2*). We hypothesize that Phi in plants stimulates the VIH1/VIH2-catalyzed synthesis of InsP$_8$, and suppresses Pi starvation responses (*Figure 6E–F*), rationalizing physiological studies (*Ticconi et al., 2001*; *Jost et al., 2015*).

Taken together, our findings support a signaling cascade in which InsP$_8$ is generated and broken down by VIH1/VIH2 in response to changing concentrations of ATP and Pi, to maintain Pi homeostasis and to trigger Pi starvation responses via their SPX domain receptors. Future studies will be required to test this hypothesis in biochemical, mechanistic and physiological terms.

## Materials and methods

### Plant material and growth conditions

Plants were grown at 21°C, 50% humidity and in a 16 hr light: 8 hr dark cycle. For root imaging, western blots and RT-PCR, seedlings were grown on vertical plates containing half-strength

Murashige and Skoog ($^{1/2}$MS, Duchefa) media containing 1 (w/v) % sucrose, and 0.8 (w/v) % agar (Duchefa) for 6 to 10 d. Seeds were sequentially surface-sterilized using 70% (v/v) ethanol, a 5% hypochlorite solution (Javel, 13–14%), and rinsed 4 times with sterilized water.

The Arabidopsis seeds of T-DNA insertion lines were obtained from the European Arabidopsis Stock Center (http://arabidopsis.info/) except for *phr1 phl1, ipk1-1,* which were kindly provided by Dr. Javier Paz-Ares (National Center for Biotechnology, Madrid, Spain) and Dr. Yves Poirier (University of Lausanne, Switzerland). The T-DNA lines used in this study are as follows: *vih1-2* (SALK_094780C), *vih1-4* (WiscDsLox293-296invF12), *vih1-6* (SAIL_175_H09), *vih2-4* (GK_080A07), *vih2-6* (GK_008H11), *spx1-1* (SALK_092030C), *spx2-1* (SALK_080503C), *phl1* (SAIL_731_B09), *ipk1-1* (SALK_065337). *phr1* used in this study is an EMS mutant (*Rubio et al., 2001*).

## Characterization of T-DNA insertion mutants

Homozygous lines were identified by PCR using T-DNA left and right border primers paired with gene-specific sense and antisense primers (*Supplementary file 2a*). mRNA expression levels were quantified using cDNA prepared from RNA extracts of seedlings 9 d after germination (DAG). Primers flanking the T-DNA position of the *vih1-2* allele were used in qRT-PCR quantifications of VIH1 expression for *vih1-2, vih1-4* and *vih1-6*. VIH2 expression in *vih2-4* and *vih2-6* plants was quantified using primers flanking the respective T-DNA positions. Primer sequences are listed in *Supplementary file 2d*. The crosses between different T-DNA lines and the analysis of their segregation ratios are listed in *Supplementary file 1a,b*.

## Generation of transgenic lines

All transgenic Arabidopsis lines are listed in *Supplementary file 1c*. VIH1, VIH2, SPX1 and SPX3 were amplified from Arabidopsis cDNA and introduced into either pB7m34GW or pH7m34GW binary vectors. Mutations targeting the kinase or phosphatase domains were introduced by site-directed mutagenesis (mutagenesis primers listed in *Supplementary file 2b*). VIH1 targeting amiR-NAs were designed using WMD3 – Web MicroRNA Designer (http://wmd3.weigelworld.org/). Fragments containing AtVIH1$^{amiRNA}$ were amplified and introduced into the pMDC7 binary vector using the primers listed in *Supplementary file 2b* (*Curtis and Grossniklaus, 2003*). Fragments coding for full-length HsPPIP5K2, its kinase (HsPPIP5K2-KD) or its phosphatase (HsPPIP5K2-PD) domain were amplified by PCR (primers listed in *Supplementary file 2b*) using a synthetic HsPPIP5K2 gene (codon was optimized for expressing in Arabidopsis, Geneart -Thermo Fisher Scientific) as template.

Binary vectors were assembled by the multi-site Gateway technology (Thermo Fisher Scientific). All constructs were transformed into *Agrobacterium tumefaciens* strain pGV2260. Plants were transformed using the floral dip method (*Clough and Bent, 1998*). Transformants were selected in $^{1/2}$MS medium supplemented with BASTA or Hygromycin down to T3 generation. The quadruple mutant was obtained by crossing *phr1 phl1* homozygous plants with *vih1-2 vih2-4* heterozygous plants.

## Determination of cellular Pi concentrations

To measure cellular Pi concentration at seedling stage, seedling 7 DAG were transferred from vertical plates containing $^{1/2}$MS, 1 (w/v) % sucrose, and 0.8 (w/v) % agar to new vertical plates containing Pi deficient $^{1/2}$MS (Duchefa), 1 (w/v) % sucrose, and 0.8 (w/v) % agarose (A9045, Sigma), and supplemented with varying concentrations of Pi ($KH_2PO_4/K_2HPO_4$ mixture, pH 5.7). To measure shoot Pi concentration at adult stage, seedlings 7 DAG were transplanted from vertical plates to soil. Seedlings 14 DAG and adult plants 20 DAG were dissected and Pi content was measured by the colorimetric molybdate assay (*Ames, 1966*).

## Plant protein extraction and western blotting

Around 250 mg seedlings 10 DAG were harvested and frozen in liquid nitrogen in 2 ml eppendorf tubes with metal beads and ground in a tissue lyzer (MM400, Retsch). Ground plant material was resuspended in 500 µl extraction buffer (150 mM NaCl, 50 mM Tris-HCl pH 7.5, 10% (v/v) glycerol, 1% (v/v) Triton X-100, 5 mM DTT) containing a protease inhibitor cocktail (P9599, Sigma, one tablet/20 ml), centrifuged for 30 min at 4,000 rpm and at 4°C. Protein concentrations were estimated using a Bradford protein assay. Around 100 µg protein was mixed with 2x SDS loading buffer, boiled at 95°C for 10 min and separated on 8% SDS-PAGE gels. Anti-GFP antibody coupled with horse radish

peroxidase (HRP, Miltenyi Biotec) at 1:2000 dilution was used to detect eGFP/mCit tagged protein constructs.

## RNA analyses

Around 200 mg seedlings at 14 DAG were harvested in 2 ml eppendorf tubes and frozen in liquid nitrogen. 1–2 µg total RNA was extracted using a RNeasy plant mini kit (Qiagen, Valencia, CA) and reverse-transcribed using the SuperScript VILO cDNA synthesis kit (Invitrogen, Grand Island, NY). qRT-PCR amplifications were monitored using SYBR-Green fluorescent stain (Applied Biosystems), and measured using a 7900HT Fast Real Time PCR-System from Applied Biosystems (Carlsbad, CA). For the normalization of gene transcripts, *ACTIN2* or *ACTIN8* was used as a internal control (primers listed in *Supplementary file 2d*). Averages of triplicate or quadruplicate reactions ± SE are shown. All of the quantifications were repeated at least 3 times, with similar results. Average $2^{-\Delta CT}$ values were calculated over three technical and three biological replicates. The original data are shown in *Supplementary file 3*. $2^{-\Delta CT}$ values were converted into Z-scores and plotted in heatmapper (http://heatmapper.ca) (*Babicki et al., 2016*).

## Confocal microscopy

Zeiss LSM780 equipment was used for confocal laser scanning. Confocal settings were set to record the emission of eGFP (excitation 488 nm, emission 500–550 nm), and mCit (excitation 514 nm, emission 524–580 nm). For the imaging of Arabidopsis root tips and pollen, a 40 x/1.3 oil objective and 20 x/0.50 objective were used, respectively. Images were acquired with Zen 2012 SP2 using identical settings for all samples. All image analyses were performed in the FIJI distribution of ImageJ 1.51 w (http://imagej.nih.gov/ij/).

## β-glucuronidase assay

For GUS assays, fragments encompassing 1.85 kb and 1.80 kb of the promoter regions of VIH1 and VIH2, respectively, were amplified from Arabidopsis genomic DNA by using primers listed in *Supplementary file 2b*, and cloned using the Gateway LR reaction to generate pB7m34GW binary vectors expressing pVIH1::GUS and pVIH2::GUS. Plant tissues from the transgenic lines were fixed in sodium phosphate buffer (50 mM sodium phosphate, pH 7) containing 2 (v/v) % formaldehyde for 30 min at room temperature and subsequently, washed 2 times with the sodium phosphate buffer. The fixed plant tissues were suspended in staining buffer (50 mM sodium phosphate, 0.5 mM K - ferrocyanide, 0.5 mM K - ferrocyanide, 1 mM X-GlcA) vacuum infiltrated for 15 min, and stained for 4–6 hr at 37°C. The stained plant tissues were then imaged directly or stored in 20% EtOH after two washes with 96 and 60 (v/v) % ethanol, respectively, for 1 hr. Images were acquired with a Zeiss StiREO Discovery.V8 microscope.

## Inositol polyphosphate extraction from seedlings and HPLC analyses

Extraction and measurement of inositol polyphosphates from Arabidopsis seedlings were performed as follows. Seedlings were grown under sterile conditions on solid $^{1/2}$MS media containing 1 (w/v) % sucrose and 0.8 (w/v) % agar for 8 DAG, and then transferred to $^{1/2}$MS liquid medium including 30 µCi mL$^{-1}$ of [$^3$H]-myo-inositol (30 to 80 Ci mmol$^{-1}$ and 1 mCi mL$^{-1}$; Biotrend; ART-0261–5). After 5 d of labeling, seedlings were washed two times with ultrapure water before harvesting and freezing into liquid nitrogen. InsPs were extracted as described previously (*Azevedo and Saiardi, 2006*) and resolved by strong anion exchange chromatography HPLC (using the partisphere SAX 4.6 × 125 mm column; HiChrom Ltd) at a flow rate of 0.5 ml min$^{-1}$. The column was eluted with a gradient generated by mixing buffers A (1 mM EDTA) and B [1 mM EDTA and 1.3 M (NH$_4$)$_2$HPO$_4$, pH 3.8, with H$_3$PO$_4$] with the following gradient: 0–7 min, 0% buffer B; 7–68 min, 10–84% buffer B; 68–82 min, 84–100% buffer B; 82–100 min, 100% buffer B; 100–125 min, 0% buffer B. The total time of this program is 125 min at a flow rate of 0.5 ml min$^{-1}$. Fractions 1–95 are collected and counted. Fractions were collected each minute, mixed with scintillation cocktail (Perkin-Elmer; ULTIMA-FLO AP), and analyzed by scintillation counting. To account for differences in fresh weight and extraction efficiencies between samples, values shown are normalized activities based on the total activity of each sample. 'Total' activities for normalization were calculated by counting fractions from 25 min (InsP$_3$) until the end of the run.

## Protein expression and purification

Synthetic genes coding for AtVIH1 (residues 1–1049), AtVIH2 (residues 1–1050), HsPPIP5K2 (residues 1–1124) and ScVip1 (residues 1–1146), codon optimized for expression in *Spodoptera frugiperda* Sf9 cells were obtained from Geneart (Thermo Fisher Scientific). The fragments coding for the kinase domain of AtVIH2 (residues 11–338), HsPPIP5K2 (residues 41–366), the fragments coding for the phosphatase domain of AtVIH2 (residues 359–1002), ScVip1 (residues 515–1088), and the fragments encoding both the kinase domain and the phosphatase domain of AtVIH1 (residues 8–1016), AtVIH2 (residues 1–1050) and ScVip1 (residues 188–1088) were amplified from the synthetic genes. Point mutations were introduced by site-directed mutagenesis. All DNA fragments were cloned into a modified pFastBac1 vector (Geneva Biotech), providing a N-terminal TEV (tobacco etch virus protease) cleavable Strep-9xHis tandem affinity tag (primers are listed in *Supplementary file 2c*), and all constructs were confirmed by DNA sequencing.

Bacmids were generated by transforming the plasmids into *Escherichia coli* DH10MultiBac (Geneva Biotech), isolated, and transfected into *Spodoptera frugiperda* Sf9 cells with profectin transfection reagent (AB Vector) followed by viral amplification. All proteins were expressed in *Spodoptera frugiperda* Sf9 cells and harvested from the medium 3 days post infection, and purified separately by sequential $Ni^{2+}$ (HisTrap excel HP, GE Healthcare) and Strep (Strep-Tactin XT Superflow, IBA, Germany) affinity chromatography. Next, proteins were further purified by size-exclusion chromatography on a HiLoad 26/600 Superdex 200 pg column (GE Healthcare) or on a Superdex 200 increase 10/300 GL column (GE Healthcare), equilibrated in 150 mM NaCl, 20 mM HEPES pH 7.0. Protein purity and molecular weight were analyzed by SDS - PAGE and MALDI-TOF mass spectrometry.

## NMR-based enzyme assays

For kinase assays, proteins encoding full-length proteins of ScVIP1, or the isolated kinase domains of AtVIH2 and HsPPIP5K2 were used. 1–10 µM of protein was incubated in a buffer containing 20 mM HEPES pH 7, 50 mM NaCl, 1 mM DTT, 2.5 mM ATP, 5 mM creatine phosphate, 1 U creatine kinase, 7.5 mM $MgCl_2$ and 175 µM of $[^{13}C_6]InsP_6$ or $[^{13}C_6]5PP-InsP_5$, at a final volume of 550 µl. The reaction was incubated at 37°C overnight, quenched with 50 µl 0.7 M EDTA, lyophilized and resuspended in 600 µl of 100% (v/v) $D_2O$. The samples were measured as previously described (*Harmel et al., 2019*). In brief, Bruker AV-III spectrometers (Bruker Biospin, Rheinstetten, Germany) using cryogenically cooled 5 mm TCI-triple resonance probe equipped with one-axis self-shielded gradients and operating at 600 MHz for proton nuclei, 151 MHz for carbon nuclei, and 244 MHz for phosphorous nuclei were used. The software used to control the spectrometer was topspin 3.5 pl6. Temperature was calibrated using $d_4$-methanol and the formula of *Findeisen et al. (2007)*. Kinetic assays were performed similarly with the exception that the labeled $[^{13}C_6]InsPs$ were added just before NMR measurements and were not quenched by EDTA.

For phosphatase assays using the isolated ScVip1 phosphatase domain (ScVip1-PD), reactions contained 2 µM ScVip1-PD in 20 mM HEPES pH 7.0, 150 mM NaCl, 1 mg/ml BSA, 0, 1 or 10 mM $K_2HPO_4$, respectively, and $D_2O$ (total volume 600 µl). Reactions were pre-incubated at 37°C and started by adding 40 µM $[^{13}C_6]5PP-InsP_5$. Reactions were quenched after 0, 5, 10, 20, and 40 min by boiling at 90°C for 5 min. The samples were measured by NMR with a pseudo-2D spin-echo difference experiment and the relative intensities of the C2 peaks of $[^{13}C_6]InsP_6$ and $[^{13}C_6]5PP-InsP_5$ were quantified (*Harmel et al., 2019*).

The activity of full-length ScVip1 vs. different ATP-$Mg^{2+}$ levels was assessed in reactions containing 1 µM ScVip1-FL in 20 mM HEPES pH 6.8, 150 mM NaCl, 1 mg/ml BSA, and the ATP-$Mg^{2+}$ concentration indicated (total volume 150 µl). Reactions were pre-incubated at 37°C and started by adding 80 µM $[^{13}C_6]5PP-InsP_5$. Reactions were quenched after 2, 5, 10 min by adding 20 µl of 500 mM EDTA, lyophilized, dissolved in 600 µl $D_2O$. The remaining protein was heat-inactivated for 5 min at 90°C and the precipitated protein removed by centrifugation. $^1H,^{13}C$-HMQC spectra of the samples were recorded and the relative signal intensities of the C2 peaks were quantified.

## Malachite green-based phosphatase assays

InsP_6 was purchased from SiChem, Germany. 1PP-InsP_5 (*Capolicchio et al., 2013*), 5PP-InsP_5 and the non-hydrolyzable 5-methylene-bisphosphonate inositol pentakisphosphate (5PCP-InsP_5) were

synthesized as previously described (*Wu et al., 2013*). Purified recombinant AtVIH2-PD (~8 µg), AtVIH2-PD$^{RH/AA}$ (~5 µg), ScVip1-PD (~27 µg) and HsPPIP5K2-KD (~17 µg) were incubated with 175 µM of each inositol polyphosphate substrate for 16 hr at 37°C in 20 mM HEPES pH 7.0, 150 mM NaCl, 1 mg/ml BSA. Reactions were stopped by rapid freezing in liquid nitrogen. Reaction products were separated by electrophoresis in 35% acrylamide gels in TBE buffer (89 mM Tris, pH 8.0, 89 mM boric acid, 2 mM EDTA) and stained with toluidine blue (20% [v/v] methanol, 0.1% [w/v] toluidine blue, and 3% [v/v] glycerol) as described (*Losito et al., 2009*). Phosphatase activities were determined using the malachite green phosphate assay (*Baykov et al., 1988*) kit (Sigma), following the manufacturer's instructions.

## Gel-based bifunctional kinase and phosphatase activity assays

2 µM recombinant ScVip1(188–1088) was incubated with 80 µM 5PP-InsP$_5$ for 16 hr at 37°C in a buffer containing 150 mM NaCl, 20 mM HEPEs, pH 7.0, and 1 mg / ml BSA, and variable concentrations of ATP-Mg$^{2+}$, as indicated. Reactions were stopped by rapid freezing in liquid nitrogen, and samples were subsequently separated in 35% acrylamide gels in TBE buffer and stained with toluidine blue.

## Determination of cellular ATP/ADP levels in seedlings

The contents of ADP/ATP were measured by High Performance Liquid Chromatography (HPLC) with 20 mg of fresh weight plant materials (*Liang et al., 2015*). Briefly, 0.2 ml of 0.1 M HCL was mixed with the plant materials on ice. 15 µl extract/standard (different concentration) was mixed with 77 µl CP Buffer (62 mM, citric acid monohydrate and 76 mM (Na)$_2$HPO$_4$ × 2H$_2$O) and 8 µl 45% chloroacetaldehyde and incubated at 80°C for 10 min. The mixture was then centrifuged at 16.000 xg at 20°C for 30 min. 90 µl of the supernatant was then measured by the HPLC (Hyperclone C18 (ODS) column (Phenomenex)). The result was calculated by the standard ADP and ATP gradient.

## Statistics

Simultaneous inference was used throughout to limit the false positive decision rate in these randomized one- or two-way layouts. Designs in which single plants represent the experimental unit were analyzed using a linear model, whereas for designs with technical replicates, a mixed effect model was used accordingly. Normal distributed variance homogeneous errors were assumed when appropriate, otherwise a modified variance estimator allowing group-specific variances was used (*Herberich et al., 2010*). Multiple comparisons of several genotypes vs. wild-type (Col-0) were performed according to *Dunnett (1955)* (*Figure 1F*, *Figure 1—figure supplement 5B*, *Figure 2C*, *Figure 2—figure supplement 2D*, *Figure 2—figure supplement 3B*, *Figure 3B,F,G*, *Figure 4D*, *Figure 4—figure supplement 3C*, *Figure 6F*). Tukey-type all-pairs comparisons between the genotypes (either unpooled or pooled, indicated in the respective figure panel by a horizontal line) were used for data represented in *Figure 1—figure supplement 5D,G*, *Figure 2—figure supplement 4C*, *Figure 3—figure supplement 1B*, and *Figure 6D*. Trend analyses were performed for qualitative levels by a multiple contrasts test (*Williams, 1971*) in *Figure 1E* and *Figure 3C*, and for quantitative levels a regression-type test was employed (*Tukey et al., 1985*).

All p-values are for 2-sided tests (***p<0.0001; **p<0.001; *p<0.01 for mutant vs. wild-type; ### p<0.0001; ## p<0.001; # p<0.01 for mutant vs. mutant). All computations were performed in R version 3.3.2., using the packages multcomp (*Hothorn et al., 2008*), nlme and tukeytrend (*Tukey et al., 1985*).

## Acknowledgements

This work was supported by European Research Council under the European Union's Seventh Framework Programme (FP/2007–2013)/ERC Grant Agreement 310856 (to MH), by Swiss National Foundation Sinergia Grant CRSII5_170925 (to DF and MH), by grant SCHA 1274/4–1 from the Deutsche Forschungsgemeinschaft (to GS) and by an HHMI International Research Scholar Award (to MH). KL was supported by an EMBO long-term fellowship (ALTF-493–2015). RKH and RP were supported by the Leibniz Gemeinschaft (SAW-2017-FMP-1). YZ and ARF were supported by the Max-Planck Society and the European Union's Horizon 2020 research and innovation program, project

PlantaSYST. We thank D Couto, L Lorenzo-Orts, M Ried, J Savarin and Y Poirier for critically reading the manuscript.

## Additional information

### Funding

| Funder | Grant reference number | Author |
| --- | --- | --- |
| H2020 European Research Council | 310856 | Michael Hothorn |
| Schweizerischer Nationalfonds zur Förderung der Wissenschaftlichen Forschung | CRSII5_170925 | Dorothea Fiedler Michael Hothorn |
| Howard Hughes Medical Institute | 55008733 | Michael Hothorn |
| European Molecular Biology Organization | ALTF 493-2015 | Kelvin Lau |
| Leibniz-Gemeinschaft | SAW-2017-FMP-1 | Dorothea Fiedler |
| Deutsche Forschungsgemeinschaft | SCHA 1274/4-1 | Gabriel Schaaf |
| Max-Planck-Gesellschaft | | Youjun Zhang Alisdair R Fernie |
| Horizon 2020 Framework Programme | PlantaSYST | Youjun Zhang Alisdair R Fernie |

The funders had no role in study design, data collection and interpretation, or the decision to submit the work for publication.

### Author contributions

Jinsheng Zhu, Conceptualization, Resources, Formal analysis, Validation, Investigation, Visualization, Methodology, Writing—original draft, Writing—review and editing; Kelvin Lau, Conceptualization, Formal analysis, Validation, Investigation, Visualization, Methodology, Writing—original draft, Writing—review and editing; Robert Puschmann, Robert K Harmel, Investigation, Methodology, Writing—review and editing; Youjun Zhang, Formal analysis, Methodology; Verena Pries, Formal analysis, Investigation, Methodology; Philipp Gaugler, Methodology; Larissa Broger, Investigation, Methodology; Amit K Dutta, Resources, Methodology; Henning J Jessen, Resources, Methodology, Writing—review and editing; Gabriel Schaaf, Conceptualization, Resources, Formal analysis, Funding acquisition, Investigation, Methodology; Alisdair R Fernie, Conceptualization, Formal analysis, Funding acquisition, Investigation, Methodology; Ludwig A Hothorn, Software, Formal analysis, Methodology, Writing—review and editing; Dorothea Fiedler, Conceptualization, Resources, Supervision, Funding acquisition, Investigation, Methodology, Project administration, Writing—review and editing; Michael Hothorn, Conceptualization, Supervision, Funding acquisition, Validation, Investigation, Writing—original draft, Project administration, Writing—review and editing

### Author ORCIDs

Jinsheng Zhu (iD) https://orcid.org/0000-0002-8131-1876
Robert Puschmann (iD) http://orcid.org/0000-0002-6443-2326
Michael Hothorn (iD) https://orcid.org/0000-0002-3597-5698

### Decision letter and Author response
Decision letter https://doi.org/10.7554/eLife.43582.031
Author response https://doi.org/10.7554/eLife.43582.032

## Additional files

### Supplementary files

• Supplementary file 1. Transgenic lines and genetic crosses used in the study. (a) Crossing of transgenic *Arabidopsis thaliana* VIH T-DNA lines. ♀: used as mother for crossing; ♂: used as father for crossing. D: double mutant that shows a characteristic *vih1-2 vih2-4* growth phenotype; W: double mutant that displaying a wild-type like phenotype. (b) Segregation ratio analysis in the progeny of independent lines of heterozygous plants shown in (a). (c) Overview of all transgenic lines used in this study.
DOI: https://doi.org/10.7554/eLife.43582.026

• Supplementary file 2. Primers used in this study. (a) Primers for characterizing T-DNA mutants. (b) Primers for the cloning of binary vectors. (c) Primers for cloning constructs for recombinant protein expression in insect cells. (d) Primers for q/RT-PCR experiments.
DOI: https://doi.org/10.7554/eLife.43582.027

• Supplementary file 3. Original qPCR data (expressed as $2^{-\Delta CT}$ values) used to generate expression heat maps. (a) Data corresponding to heat maps shown in *Figure 1G*. (b) Data corresponding to heat maps shown in *Figure 2D*. (c) Data corresponding to heat maps shown in *Figure 3D*. (d) Data corresponding to heat maps shown in *Figure 2—figure supplement 3C*. (e) Data corresponding to heat maps shown in *Figure 6E*.
DOI: https://doi.org/10.7554/eLife.43582.028

• Transparent reporting form
DOI: https://doi.org/10.7554/eLife.43582.029

### Data availability

Pi measurements: raw data included in actual figure. Phenotypes: representative lines shown in main figures, at least three independent lines shown in figure supplements. Western blots: full western blots shown in figure supplements. Protein gels: Full gels shown in figure 5— figure supplement 1 and figure 6—figure supplement 1. DNA sequences of the truncated VIH2 transcript can be found in figure 2— figure supplement 1. NMR data: full 1D and 2D spectra shown in figure 5 and figure 5— figure supplement 2, figure 6—figure supplement 2.

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
