## [Decision Letter]

Thank you for submitting your article "Two bifunctional inositol pyrophosphate kinases/phosphatases control plant phosphate homeostasis" for consideration by *eLife*. Your article has been reviewed by two peer reviewers, and the evaluation has been overseen by a Reviewing Editor and Ian Baldwin as the Senior Editor. The reviewers have opted to remain anonymous.

The reviewers have discussed the reviews with one another and the Reviewing Editor has drafted this decision to help you prepare a revised submission. Both reviewers were very positive about your work on phosphate homeostasis, but also were reserved about the bifunctional mode of action. They outlined several suggestions (see reviewer's comments below), which may guide you to further strengthen your nice manuscript. The reviewers came to a constructive consensus, which we would like you to experimentally assess.

Essential revisions:

1) Both reviewers came to the conclusion that the molecular analysis of the Pi starvation response would have been greatly benefited from the examination of a large set of Pi starvation responsive genes.

2) Moreover, they believe it is essential to perform in vitro analysis of the impact of phosphate and phosphite on the kinase and phosphatase activity of the VIH proteins. In detail they suggest to examine ATP levels after Pi starvation and after re-feeding the starved plants with phosphite (and phosphate as control). Re-feeding could be for 8, 24 and 48 hours. In the analysis, in addition to ATP, Pi levels and plant biomass should be measured, and the expression of Pi starvation induced genes should be tested.

3) The reviewers also suggested to use in vitro analysis to compare ksc1 with the vih1/vih2 double mutant. Instead of expanding all figures, the reviewers proposed to use the before suggested in vitro experiment also to compare ksc1 with the double mutant.

4) Introducing SPX3 or SPX1 fusion with e-GFP in the vih1/vih2 double mutant background will not be feasible within two months (or only in transient assay, which would be less reliable). Hence, the reviewers did not insist, but encouraged to include such data, if available.

*Reviewer #1:*

In this very interesting manuscript the authors provide numerous data supporting a key role of PP-Ins to control Pi homeostasis. In a previous article they demonstrated that PP-Ins could bind SPX regulatory domain. Here by genetic manipulations of two key enzymes (VIH1 and VIH2) involved in the biosynthesis in planta of these components they promoted important physiological effect related to Pi content and Pi homeostasis.

Results are of high quality but very dense and sometime confusing due to the impressive amount of data provided. In particular theses enzymes turn out to be putatively bifunctional exhibiting kinase and phosphatase activities. If the authors could clearly demonstrate the importance of the kinase activity, the results on the phosphatase part remains far to be so clear. I would therefore suggest to reduce part of the Results concerning them to generate a more fluid manuscript. I am also surprised that SPX3 or SPX1 fusion with e-GFP (used here as control in some experiments) have not been introduced in the vih1/vih2 double mutant background to investigate the impact of manipulating PP InsP on their putative direct targets.

*Reviewer #2:*

This study concerns the functional analysis of two bifunctional inositol pyrophosphate kinases/phosphatases from Arabidopsis, VIH1 and VIH2 revealing their role in Pi homeostasis. With the contentions mentioned below, this is an important finding because it adds support to the emerging idea that inositol pyrophosphates act as signal molecules in Pi homeostasis. Specifically, here the authors show that the double mutant vih1vih2 displays constitutive activation of Pi starvation responses leading to severe growth impairment. In fact, inactivation of PHR1 and PHL1 master regulators of Pi starvation responses greatly rescue vih1vih2 phenotype. The authors also analyzed site-specific mutants of vih1 and 2 at the kinase and phosphatase domains and find they have opposing effects on Pi accumulation and on expression of Pi starvation induced genes. Finally, biochemical analyses in vitro show the levels of ATP determine the relative kinase/phosphatase activity of these bifunctional enzymes. Since ATP levels have been shown to drop under Pi starvation, it has been argued ATP could help to translate to inositol polyphosphate the information on Pi levels. A remaining issue is to examine whether Pi can directly impact these enzymatic activities (as it was shown for the animal counterparts, Gu et al., 2017). This is important because this provides a direct mechanism of translation to insPP of information on P levels). In addition, it could help to explain some longstanding observations in plant physiology regarding the inhibition of Pi starvation responses by phosphite, a non-metabolizable analog of Pi).

Overall, in this reviewer's opinion the study is technically sound and potentially interesting but the analysis of the impact of Pi (and phosphite) on VIH in vitro activity is necessary to complete the story. In line with this, it is also important to examine whether phosphite affect ATP levels in vivo.

---

## [Author Response]

Essential revisions:1) Both reviewers came to the conclusion that the molecular analysis of the Pi starvation response would have been greatly benefited from the examination of a large set of Pi starvation responsive genes.

As requested, we have re-performed all qPCR analyses, now using a set of ten validated PSI marker genes (and two non-PSI genes PHR1 and LPR1). The target genes were selected following the advice of Laurent Nussaume (CEA, F), an expert on plant Pi starvation responses. These new experiments are now shown as heat maps in revised Figure 1G (previously Figure 2), Figure 2E (previously Figure 3), Figure 2—figure supplement 3, Figure 3D (previously Figure 4), and Figure 6—figure supplement 2.

2) Moreover, they believe it is essential to perform in vitro analysis of the impact of phosphate and phosphite on the kinase and phosphatase activity of the VIH proteins. In detail they suggest to examine ATP levels after Pi starvation and after re-feeding the starved plants with phosphite (and phosphate as control). Re-feeding could be for 8, 24 and 48 hours. In the analysis, in addition to ATP, Pi levels and plant biomass should be measured, and the expression of Pi starvation induced genes should be tested.

We have performed all the suggested experiments, which now enables us to clearly link cellular ATP and Pi levels to the VIH1/2-mediated control of cellular InsP_8_ concentrations:

We now show that phosphite stimulates InsP_8_ synthesis by PPIP5K to a similar extend than Pi (newly added Figure 6—figure supplement 2B). The respective paragraph in the Results section reads: “To rationalize the puzzling finding that Phi and Pi can both suppress Pi starvation responses, yet only Pi can promote physiological changes, we next tested if Pi and/or Phi would have an effect on PPIP5K enzyme activity. […] In physiologically relevant growth conditions, Pi and ATP/ADP levels may together shape the output of the Pi starvation response, by controlling the synthesis and breakdown of the InsP_8_ signaling molecule generated by VIH1 and VIH2.”

The Discussion section has been changed accordingly: “Using the isolated phosphatase and full-length ScVip1, we further characterized Pi as an inhibitor of the phosphatase activity, while stimulating the kinase activity in the context of the full-length enzyme (Figure 6—figure supplement 2). […] We hypothesize that Phi in plants stimulates the VIH1/VIH2-catalyzed synthesis InsP_8_, and suppresses Pi starvation responses (Figure 6E-F), rationalizing physiological studies (Ticconi et al., 2001; Jost et al., 2015).”

We have quantified ATP/ADP levels in the suggested time course experiment (8, 24, 48h) in seedlings grown in Pi deficient medium, or after re-feeding with 0.5 mM Pi or Phi, respectively (newly added Figure 6C). We find: “… that ATP/ADP ratios increased after Pi refeeding over a 48h time course, and were significantly higher when compared to the 0 mM Pi control. Importantly, this was not the case when we used 0.5 mM Phi instead of Pi in the experiment (Figure 6D)” (newly added Figure 6D).

As suggested by the reviewers, we also report Pi levels, seedling root length, seedling fresh weight and PSI gene expression for all conditions in this experiment (newly added Figure 6E, F). The respective paragraph in the Results reads: “We confirmed that the conditions chosen for this experiment were physiological relevant and found that upon Pi re-feeding plants showed significantly increased fresh weight, root length and cellular Pi content (Figure 6F). […] No such effect was observed when using 0.5 mM Phi instead of Pi (Figure 6C, F), despite both molecules suppressing PSI gene expression (Figure 6E).”

We have added two statements to our Discussion: “We found cellular ATP levels to be lower in Arabidopsis seedling grown under Pi deficient when compared to Pi replete conditions (Figure 6D), in good agreement with what has been reported in other plant species, and in other organisms (Duff et al., 1989; Choi et al., 2017; Gu et al., 2017a).” and: “Importantly, while transfer of seedlings to Phi containing media does not induce ATP level changes, or physiological responses (Figure 6D, F), the Phi re-feeding suppresses PSI starvation responses and PSI target gene expression (Figure 6E), as previously shown (Ticconi et al., 2001; Jost et al., 2015).”

3) The reviewers also suggested to use in vitro analysis to compare ksc1 with the vih1/vih2 double mutant. Instead of expanding all figures, the reviewers proposed to use the before suggested in vitro experiment also to compare ksc1 with the double mutant.

We now have included a side by side comparison of the previously characterized *ipk1-1* mutant (Kue et al., Plant J, 2014) with our vih1-2 vih2-4 single and double mutants. The vitro analysis suggested by the reviewers has been performed using seedlings 14 DAG grown in Pifree media, supplemented with either 0, 1 or 10 mM Pi. Growth phenotypes are shown in revised Figure 1D, quantification in 1E. The respective Pi contents are shown in Figure 1F and a comparative PSI gene expression analysis is shown in Figure 1G. From this comparison, we found that *vih1-2 vih2-4* shows a much stronger growth phenotype compared to *ipk1-1*. Both mutants shown increased cellular Pi levels with *vih1-2 vih2-4* accumulating significantly more Pi when compared to the *ipk1-1* and the wild-type control. PSI gene expression is most strongly misregulated in *vih1-2 vih2-4*, as shown in Figure 1G.

4) Introducing SPX3 or SPX1 fusion with e-GFP in the vih1/vih2 double mutant background will not be feasible within two months (or only in transient assay, which would be less reliable). Hence, the reviewers did not insist, but encouraged to include such data, if available.

During the revision of our manuscript, we have generated the suggested crosses. Specifically, we crossed pSPX1::SPX1-eGFP, pSPX3::SPX3-eGFP and pPHR1:eGFP-PHR1 lines into our seedling lethal *vih1-2 vih2-4* background. We now show in newly added Figure 4—figure supplement 2 that under Pi replete conditions (~0.7 mM), that the PSI marker proteins SPX1 and SPX3 accumulate in the double mutant, but not in wild-type, while the non PSI gene product PHR1 shows not such behavior. Since these lines were only recently obtained, we did not perform a full physiological characterization in different Pi regimes thus far, which we hope will be acceptable to reviewer #1. We have added a statement in the Results section of our manuscript that reads: “We next introduced pSPX1::SPX1-eGFP and pSPX3::SPX3-eGFP into our *vih1-2 vih24* double mutant, and found the PSI marker proteins to be highly abundant in the double mutant background when expressed under Pi repleted conditions (Figure 4—figure supplement 2). A pPHR1::eGFP-PHR1 control line did not show this behavior (Figure 4—figure supplement 2).”

Reviewer #1:

In this very interesting manuscript the authors provide numerous data supporting a key role of PP-Ins to control Pi homeostasis. In a previous article they demonstrated that PP-Ins could bind SPX regulatory domain. Here by genetic manipulations of two key enzymes (VIH1 and VIH2) involved in the biosynthesis in planta of these components they promoted important physiological effect related to Pi content and Pi homeostasis.Results are of high quality but very dense and sometime confusing due to the impressive amount of data provided. In particular theses enzymes turn out to be putatively bifunctional exhibiting kinase and phosphatase activities. If the authors could clearly demonstrate the importance of the kinase activity, the results on the phosphatase part remains far to be so clear. I would therefore suggest to reduce part of the Results concerning them to generate a more fluid manuscript.

PPIP5Ks are bifunctional enzymes not only in plants but also in fungi and animals, where the regulatory function of the phosphatase domain has been recently appreciated (Guo et al., 2017a, Pascual-Ortiz et al., 2018). We now provide new experimental evidence for the functional relevance of the phosphatase domains: We now show in vitro enzyme assays that the catalytic activity of the phosphatase domain is directly regulated by Pi (Figure 6—figure supplement 2A). Also, we have confirmed using quantitative kinetic NMR assays that the RH/AA mutation is indeed still catalytically active (newly added Figure 5F). We strongly feel that this data should be included in the manuscript, as these mutations have not only been used by us, but previously as catalytic dead mutations by many labs studying PPIP5Ks in fungi and animals.

We have however followed the advice of the reviewer to clarify our experiments on the phosphatase domain and to tone down our discussion. The results are presented in revised Figures 2A-D, Figure 2—figure supplement 3. The respective Results statement reads: “We found that VIH2^KD/AA^ seedlings showed significantly increased cellular Pi levels, while VIH2^RH/AA^ contained less Pi when grown in Pi sufficient conditions (Figure 2—figure supplement 3). […] The Pi content of the VIH1^RH/AA^ and VIH2^RH/AA^ lines appeared however similar to the Col-0 control when plants were grown in soil (Figure 2—figure supplement 4).”

The revised Discussion reads: “In any case, we do observe statistically significant differences in Pi shoot content only in VIH2^RH/AA^ complementation lines at adult stage, with young seedling behaving similar to wild-type (Figure 2, Figure 2—figure supplement 2). […] However, the strict conservation of the bifunctionality in PPIP5Ks and the fact that we could rescue our *vih1-2 vih2-4* double mutant only with a phosphatase-mutated version of human PPIP5K2 (Figure 4—figure supplement 3) together suggest a potential regulatory role for the phosphatase domain, as previously characterized for the human and yeast enzymes (Gu et al., 2017a; Pascual-Ortiz et al., 2018).”

I am also surprised that SPX3 or SPX1 fusion with e-GFP (used here as control in some experiments) have not been introduced in the vih1/vih2 double mutant background to investigate the impact of manipulating PP InsP on their putative direct targets.

(Also see editorial comments above) During the revision of our manuscript, we have crossed pSPX1::SPX1-eGFP, pSPX3::SPX3-eGFP and pPHR1:eGFP-PHR1 lines into our seedling lethal *vih1-2 vih2-4* background. We now show in newly added Figure 4—figure supplement 2 that under Pi replete conditions (~0.7 mM), that the PSI marker proteins SPX1 and SPX3 accumulate in the double mutant, but not in wild-type, while the non PSI gene product PHR1 shows not such behavior. Since these lines were only recently obtained, we did not perform a full physiological characterization in different Pi regimes thus far, which we hope will be acceptable to reviewer #1. We have added a statement in the Results section of our manuscript that reads: “We next introduced the pSPX1::SPX1-eGFP and pSPX3::SPX3-eGFP into our *vih1-2 vih2-4* double mutant, and found the PSI marker proteins to be highly abundant in the double mutant background when expressed under Pi replete conditions (Figure 4—figure supplement 2). A pPHR1::eGFPPHR1 control line did not show this behavior (Figure 4—figure supplement 2).”

Reviewer #2:

*This study concerns the functional analysis of two bifunctional inositol pyrophosphate kinases/phosphatases from Arabidopsis, VIH1 and VIH2 revealing their role in Pi homeostasis. With the contentions mentioned below, this is an important finding because it adds support to the emerging idea that inositol pyrophosphates act as signal molecules in Pi homeostasis. Specifically, here the authors show that the double mutant vih1vih2 displays constitutive activation of Pi starvation responses leading to severe growth impairment. In fact, inactivation of PHR1 and PHL1 master regulators of Pi starvation responses greatly rescue vih1vih2 phenotype. The authors also analyzed site-specific mutants of vih1 and 2 at the kinase and phosphatase domains and find they have opposing effects on Pi accumulation and on expression of Pi starvation induced genes. Finally, biochemical analyses* in vitro *show the levels of ATP determine the relative kinase/phosphatase activity of these bifunctional enzymes. Since ATP levels have been shown to drop under Pi starvation, it has been argued ATP could help to translate to inositol polyphosphate the information on Pi levels.*

We have performed the suggested experiments in collaboration with Alisdair Fernie, Max Planck Institute for Molecular Plant Physiology, Golm and can now show that ATP levels are significantly reduced under Pi starvation (revised Figure 6D, please see above).

A remaining issue is to examine whether Pi can directly impact these enzymatic activities (as it was shown for the animal counterparts, Gu et al., 2017). This is important because this provides a direct mechanism of translation to insPP of information on P levels).

Thank you for suggesting this interesting experiment. We now present quantitative enzymatic assays by NMR that revealed that indeed Pi can inhibit the phosphatase activity (Figure 6—figure supplement 2A). Using a full-length PPIP5K we can now demonstrate that Pi promotes InsP_8_ synthesis (Figure 6—figure supplement 2B), which based on our in planta HPLC analyses represents the active signaling in Pi homeostasis and starvation responses (newly added Figure 3E, G). We have added these findings to our Abstract (“Pi inhibits the phosphatase activity of the enzyme”), Results: “We found that 5PP-InsP_5_ hydrolysis by the isolated ScVip1 phosphatase domain is inhibited in the presence of 10 mM Pi in quantitative NMR time course experiments (Figure 6—figure supplement 2). […] We found that addition of Pi or Phi promoted the synthesis of InsP_8_, which based on our in vivo and in vitro analyses represents the active signaling molecule in plant Pi homeostasis and starvation responses.” and Discussion section: “Using the isolated phosphatase and full-length ScVip1, we further characterized Pi as an inhibitor of the phosphatase activity, while promoting InsP_8_ synthesis in the context of the full-length enzyme (Figure 6—figure supplement 2).”

In addition, it could help to explain some longstanding observations in plant physiology regarding the inhibition of Pi starvation responses by phosphite, a non-metabolizable analog of Pi).

Indeed, we found that also Phi can stimulate InsP_8_ synthesis by PPIP5Ks (newly added Figure 6—figure supplement 2B). The revised Results section reads: “This prompted us to analyze the enzymatic reaction profiles of full-length ScVip1 in the presence of 0 and 10 mM Pi, or in the presence of 10 mM Phi. […] Thus, Phi may suppress PSI gene expression by stimulating the kinase activities of VIH1 and VIH2, even though the cellular ATP/ADP ratio remains relatively constant (Figure 6D).”

We have added a statement in the Discussion: “Importantly, while transfer of seedlings to Phi containing media does not induce ATP level changes, or physiological responses (Figure 6D, F), the Phi re-feeding suppresses PSI starvation responses and PSI target gene expression (Figure 6E), as previously shown (Ticconi et al., 2001; Jost et al., 2015). […] We hypothesize that Phi in plants stimulates the VIH1/VIH2 catalyzed synthesis of InsP_8_, and suppresses Pi starvation responses (Figure 6E-F), rationalizing physiological studies (Ticconi et al., 2001; Jost et al., 2015).”

Overall, in this reviewer's opinion the study is technically sound and potentially interesting but the analysis of the impact of Pi (and phosphite) on VIH in vitro activity is necessary to complete the story. In line with this, it is also important to examine whether phosphite affect ATP levels in vivo.

We hope that our newly added physiological and enzymatic assays clarify the regulatory roles of Pi and Phi (Figure 6).